# Subgroup Generalization and Fairness of Graph Neural Networks

**Jiaqi Ma** [*†]
jiaqima@umich.edu

**Junwei Deng** [*†]
junweid@umich.edu

**Qiaozhu Mei**[*‡]
qmei@umich.edu

## Abstract

Despite enormous successful applications of graph neural networks (GNNs), theoretical understanding of their generalization ability, especially for node-level tasks where data are not independent and identically-distributed (IID), has been sparse. The theoretical investigation of the generalization performance is beneficial for understanding fundamental issues (such as fairness) of GNN models and designing better learning methods. In this paper, we present a novel PAC-Bayesian analysis for GNNs under a non-IID semi-supervised learning setup. Moreover, we analyze the generalization performances on different subgroups of unlabeled nodes, which allows us to further study an accuracy-(dis)parity-style (un)fairness of GNNs from a theoretical perspective. Under reasonable assumptions, we demonstrate that the distance between a test subgroup and the training set can be a key factor affecting the GNN performance on that subgroup, which calls special attention to the training node selection for fair learning. Experiments across multiple GNN models and datasets support our theoretical results[4].

## 1 Introduction

Graph Neural Networks (GNNs) [13, 35, 20] are a family of machine learning models that can be used to model non-Euclidean data as well as inter-related samples in a flexible way. In recent years, there have been enormous successful applications of GNNs in various areas, such as drug discovery [18], computer vision [29], transportation forecasting [49], recommender systems [48], etc. Depending on the type of prediction target, the application tasks can be roughly categorized into node-level, edge-level, subgraph-level, and graph-level tasks [46].

In contrast to the marked empirical success, theoretical understanding of the generalization ability of GNNs has been rather limited. Among the existing literature, some studies [9, 11, 25] focus on the analysis of graph-level tasks where each sample is an entire graph and the samples of graphs are IID. A very limited number of studies [36, 42] explore GNN generalization for node-level tasks but they assume the nodes (and their associated neighborhoods) are IID samples, which does not align with the commonly seen graph-based semi-supervised learning setups. Baranwal et al. [3] investigate GNN generalization without IID assumptions but under a specific data generating mechanism.

In this work, our first contribution is to provide a novel PAC-Bayesian analysis for the generalization ability of GNNs on node-level tasks with non-IID assumptions. In particular, we assume the node features are fixed and the node labels are independently sampled from distributions conditioned on the node features. We also assume the training set and the test set can be chosen as arbitrary subsets of nodes on the graph. We first prove two general PAC-Bayesian generalization bounds (Theorem 1

---

[*]School of Information, University of Michigan, Ann Arbor, Michigan, USA

[†]Equal contribution.

[‡]Department of EECS, University of Michigan, Ann Arbor, Michigan, USA

[4]Code available at `https://github.com/TheaperDeng/GNN-Generalization-Fairness`.

35th Conference on Neural Information Processing Systems (NeurIPS 2021).

and Theorem 2) under this non-IID setup. Subsequently, we derive a generalization bound for GNN (Theorem 3) in terms of characteristics of the GNN models and the node features.

Notably, the generalization bound for GNN is influenced by the distance between the test nodes and the training nodes in terms of their aggregated node features. This suggests that, given a fixed training set, test nodes that are "far away" from all the training nodes may suffer from larger generalization errors. Based on this analysis, our second contribution is the discovering of a type of unfairness that arises from theoretically predictable accuracy disparity across some subgroups of test nodes. We further conduct a empirical study that investigates the prediction accuracy of four popular GNN models on different subgroups of test nodes. The results on multiple benchmark datasets indicate that there is indeed a significant disparity in test accuracy among these subgroups.

We summarize the contributions of this work as follows:

(1) We establish a novel PAC-Bayesian analysis for graph-based semi-supervised learning with non-IID assumptions.

(2) Under this setup, we derive a generalization bound for GNNs that can be applied to an arbitrary subgroup of test nodes.

(3) As an implication of the generalization bound, we predict that there would be an unfairness of GNN predictions that arises from accuracy disparity across subgroups of test nodes.

(4) We empirically verify the existence of accuracy disparity of popular GNN models on multiple benchmark datasets, as predicted by our theoretical analysis.

## 2 Related Work

### 2.1 Generalization of Graph Neural Networks

The majority of existing literature that aims to develop theoretical understandings of GNNs have focused on the expressive power of GNNs (see Sato [34] for a survey along this line), while the number of studies trying to understand the generalizability of GNNs is rather limited. Among them, some [9, 11, 25] focus on graph-level tasks, the analyses of which cannot be easily applied to node-level tasks. As far as we know, Scarselli et al. [36], Verma and Zhang [42], and Baranwal et al. [3] are the only existing studies investigating the generalization of GNNs on node-level tasks, even though node-level tasks are more common in reality. Scarselli et al. [36] present an upper bound of the VC-dimension of GNNs; Verma and Zhang [42] derive a stability-based generalization bound for a single-layer GCN [20] model. Yet, both Scarselli et al. [36] and Verma and Zhang [42] (implicitly) assume that the training nodes are IID samples from a certain distribution, which does not align with the common practice of node-level semi-supervised learning. Baranwal et al. [3] investigate the generalization of graph convolution under a specific data generating mechanism (i.e., the contextual stochastic block model [8]). Our work presents the first generalization analysis of GNNs for non-IID node-level tasks without strong assumptions on the data generating mechanism.

### 2.2 Fairness of Machine Learning on Graphs

The fairness issues of machine learning on graphs start to receive research attention recently. Following conventional machine learning fairness literature, the majority of previous work along this line [1, 5–7, 22, 32, 39, 50] concerns about fairness with respect to a given sensitive attribute, such as gender or race, which defines protected groups. In practice, the fairness issues of learning on graphs are much more complicated due to the asymmetric nature of the graph-structured data. However, only a few studies [19] investigate the unfairness caused by the graph structure without knowing a sensitive feature. Moreover, in a node-level semi-supervised learning task, the non-IID sampling of training nodes brings additional uncertainty to the fairness of the learned models. This work is the first to present a learning theoretic analysis under this setup, which in turn suggests how the graph structure and the selection of training nodes may influence the fairness of machine learning on graphs.

### 2.3 PAC-Bayesian Analysis

PAC-Bayesian analysis [27] has become one of the most powerful theoretical framework to analyze the generalization ability of machine learning models. We will briefly introduce the background in

Section 3.2, and refer the readers to a recent tutorial [14] for a systematic overview of PAC-Bayesian analysis. We note that Liao et al. [25] recently present a PAC-Bayesian generalization bound for GNNs on IID graph-level tasks. Both Liao et al. [25] and this work utilize results from Neyshabur et al. [30], a PAC-Bayesian analysis for ReLU-activated neural networks, in part of our proofs. Compared to Neyshabur et al. [30], the key contribution of Liao et al. [25] is the derivation of perturbation bounds of two types of GNN architectures; while the key contribution of this work is the novel analysis under the setup of non-IID node-level tasks. There is also an existing work of PAC-Bayesian analysis for transductive semi-supervised learning [4]. But it is different from our problem setup and, in particular, it cannot be used to analyze the generalization on subgroups.

## 3 Preliminaries

In this section, we first formulate the problem of node-level semi-supervised learning. We also provide a brief introduction of the PAC-Bayesian framework.

### 3.1 The Problem Formulation and Notations

**Semi-supervised node classification.** Let $G = (V, E) \in \mathcal{G}_N$ be an undirected graph, with $V = \{1, 2, \ldots, N\}$ being the set of $N$ nodes and $E \subseteq V \times V$ being the set of edges. And $\mathcal{G}_N$ is the space of all undirected graphs with $N$ nodes. The nodes are associated with node features $X \in \mathbb{R}^{N \times D}$ and node labels $y \in \{1, 2, \ldots, K\}^N$.

In this work, we focus on the transductive node classification setting [47], where the node features $X$ and the graph $G$ are observed prior to learning, and every quantity of interest in the analysis will be conditioned on $X$ and $G$. Without loss of generality, we treat $X$ and $G$ as fixed throughout our analysis, and the randomness comes from the labels $y$. In particular, we assume that for each node $i \in V$, its label $y_i$ is generated from an unknown conditional distribution $\Pr(y_i \mid Z_i)$, where $Z = g(X, G)$ and $g : \mathbb{R}^{N \times D} \times \mathcal{G}_N \to \mathbb{R}^{N \times D'}$ is an aggregation function that aggregates the features over (multi-hop) local neighborhoods[5]. We also assume that the node labels are generated independently conditional on their respective aggregated features $Z_i$'s.

Given a small set of the labeled nodes, $V_0 \subseteq V$, the task of node-level semi-supervised learning is to learn a classifier $h : \mathbb{R}^{N \times D} \times \mathcal{G}_N \to \mathbb{R}^{N \times K}$ from a function family $\mathcal{H}$ and perform it on the remaining unlabeled nodes. Given a classifier $h$, the classification for a node $i$ is obtained by

$$\hat{y}_i = \operatorname*{argmax}_{k \in \{1, \ldots, K\}} h_i(X, G)[k],$$

where $h_i(X, G)$ is the $i$-th row of $h(X, G)$ and $h_i(X, G)[k]$ refers to the $k$-th element of $h_i(X, G)$.

**Subgroups.** In Section 4, we will present an analysis of the GNN generalization performance on any subgroup of the set of unlabeled nodes, $V \setminus V_0$. Note that the analysis on any subgroup is a stronger result than that on the entire unlabeled set, as any set is a subset of itself. Later we will show that the analysis on subgroups (rather than on the entire set) further allows us to investigate the accuracy disparity across subgroups. We denote a collection of subgroups of interest as $V_1, V_2, \ldots, V_M \subseteq V \setminus V_0$. In practice, a subgroup can be defined based on an attribute of the nodes (e.g., a gender group), certain graph-based properties, or an arbitrary partition of the nodes. We also define the size of each subgroup as $N_m := |V_m|, m = 0, \ldots, M$.

**Margin loss on each subgroup.** Now we can define the *empirical* and *expected margin loss* of any classifier $h \in \mathcal{H}$ on each subgroup $V_m, m = 0, 1, \ldots, M$. Given a sample of observed node labels $y_i$'s, the empirical margin loss of $h$ on $V_m$ for a margin $\gamma \geq 0$ is defined as

$$\widehat{\mathcal{L}}_m^\gamma(h) := \frac{1}{N_m} \sum_{i \in V_m} \mathbb{1}\left[h_i(X, G)[y_i] \leq \gamma + \max_{k \neq y_i} h_i(X, G)[k]\right], \tag{1}$$

where $\mathbb{1}[\cdot]$ is the indicator function. The expected margin loss is the expectation of Eq. (1), i.e.,

$$\mathcal{L}_m^\gamma(h) := \mathbb{E}_{y_i \sim \Pr(y|Z_i), i \in V_m} \widehat{\mathcal{L}}_m^\gamma(h). \tag{2}$$

---

[5]A simple example is $g_i(X, G) = \frac{1}{|\mathcal{N}(i)|+1}\left(X_i + \sum_{j \in \mathcal{N}(i)} X_j\right)$, where $g_i(X, G)$ is the $i$-th row of $g(X, G)$ and $\mathcal{N}(i) := \{j \mid (i, j) \in E\}$ is the set of 1-hop neighbors of node $i$.

To simplify the notation, we define $y^m := \{y_i\}_{i \in V_m}, \forall m = 0, \ldots, M$, so that Eq. (2) can be written as $\mathcal{L}_m^\gamma(h) = \mathbb{E}_{y^m} \widehat{\mathcal{L}}_m^\gamma(h)$. We note that the classification *risk* and *empirical risk* of $h$ on $V_m$ are respectively equal to $\mathcal{L}_m^0(h)$ and $\widehat{\mathcal{L}}_m^0(h)$.

## 3.2 The PAC-Bayesian Framework

The PAC-Bayesian framework [27] is an approach to analyze the generalization ability of a stochastic predictor drawn from a distribution $Q$ over the predictor family $\mathcal{H}$ that is learned from the training data. For any stochastic classifier distribution $Q$ and $m = 0, \ldots, M$, slightly overloading the notation, we denote the empirical margin loss of $Q$ on $V_m$ as $\widehat{\mathcal{L}}_m^\gamma(Q)$, and the corresponding expected margin loss as $\mathcal{L}_m^\gamma(Q)$. And they are given by

$$\widehat{\mathcal{L}}_m^\gamma(Q) := \mathbb{E}_{h \sim Q} \widehat{\mathcal{L}}_m^\gamma(h), \quad \mathcal{L}_m^\gamma(Q) := \mathbb{E}_{h \sim Q} \mathcal{L}_m^\gamma(h).$$

In general, a PAC-Bayesian analysis aims to bound the generalization gap between $\mathcal{L}_m^\gamma(Q)$ and $\widehat{\mathcal{L}}_m^\gamma(Q)$. The analysis is usually done by first proving that, for any "prior" distribution[6] $P$ over $\mathcal{H}$ that is independent of the training data, the generalization gap can be controlled by the discrepancy between $P$ and $Q$; the analysis is then followed by careful choices of $P$ to get concrete upper bounds of the generalization gap. While the PAC-Bayesian framework is built on top of stochastic predictors, there exist standard techniques [23] that can be used to derive generalization bounds for deterministic predictors from PAC-Bayesian bounds.

Finally, we denote the *Kullback-Leibler (KL) divergence* as $D_{\mathrm{KL}}(Q\|P) := \int \ln \frac{dQ}{dP} dQ$, which will be used in the following analysis.

# 4 The Generalization Bound and Its Implications for Fairness

As we mentioned in Section 2.3, existing PAC-Bayesian analyses cannot be directly applied to the non-IID semi-supervised learning setup where we care about the generalization (and its disparity) across different subgroups of the unlabeled samples. In this section, we first present general PAC-Bayesian theorems for subgroup generalization under our problem setup; then we derive a generalization bound for GNNs and discuss fairness implications of the bound.

## 4.1 General PAC-Bayesian Theorems for Subgroup Generalization

**Stochastic classifier bound.** We first present the general PAC-Bayesian theorem (Theorem 1) for subgroup generalization of stochastic classifiers. The generalization bound depends on a notion of *expected loss discrepancy* between two subgroups as defined below.

**Definition 1** (Expected Loss Discrepancy). *Given a distribution $P$ over $\mathcal{H}$, for any $\lambda > 0$ and $\gamma \geq 0$, for any two subgroups $V_m$ and $V_{m'}$ ($0 \leq m, m' \leq M$), define the expected loss discrepancy between $V_m$ and $V_0$ with respect to $(P, \gamma, \lambda)$ as*

$$D_{m,m'}^\gamma(P; \lambda) := \ln \mathbb{E}_{h \sim P} e^{\lambda\left(\mathcal{L}_m^{\gamma/2}(h) - \mathcal{L}_{m'}^\gamma(h)\right)},$$

*where $\mathcal{L}_m^{\gamma/2}(h)$ and $\mathcal{L}_{m'}^\gamma(h)$ follow the definition of Eq. (2).*

Intuitively, $D_{m,m'}^\gamma(P; \lambda)$ captures the difference of the expected loss between $V_m$ and $V_{m'}$ in an average sense (over $P$). Note that $D_{m,m'}^\gamma(P; \lambda)$ is asymmetric in terms of $V_m$ and $V_{m'}$, and can be negative if the loss on $V_m$ is mostly smaller than that on $V_{m'}$.

For stochastic classifiers, we have the following Theorem 1. Proof can be found in Appendix A.1.

**Theorem 1** (Subgroup Generalization of Stochastic Classifiers). *For any $0 < m \leq M$, for any $\lambda > 0$ and $\gamma \geq 0$, for any "prior" distribution $P$ on $\mathcal{H}$ that is independent of the training data on $V_0$, with*

---

[6]The distribution is called "prior" in the sense that it doesn't depend on training data. "Prior" and "posterior" in PAC-Bayesian are different with those in conventional Bayesian statistics. See Guedj [14] for details.

*probability at least $1 - \delta$ over the sample of $y^0$, for any $Q$ on $\mathcal{H}$, we have*[7]

$$\mathcal{L}_m^{\gamma/2}(Q) \leq \widehat{\mathcal{L}}_0^\gamma(Q) + \frac{1}{\lambda}\left(D_{\mathrm{KL}}(Q\|P) + \ln\frac{1}{\delta} + \frac{\lambda^2}{4N_0} + D_{m,0}^\gamma(P;\lambda)\right). \tag{3}$$

Theorem 1 can be viewed as an adaptation of a result by Alquier et al. [2] from the IID supervised setting to our non-IID semi-supervised setting. The terms $D_{\mathrm{KL}}(Q\|P), \ln\frac{2}{\delta}$, and $\frac{\lambda^2}{4N_0}$ are commonly seen in PAC-Bayesian analysis for IID supervised setting. In particular, when setting $\lambda = \Theta(\sqrt{N_0}), \frac{1}{\lambda}\left(\ln\frac{2}{\delta} + \frac{\lambda^2}{4N_0}\right)$ vanishes as the training size $N_0$ grows. The divergence between $Q$ and $P$, $D_{\mathrm{KL}}(Q\|P)$, is usually considered as a measurement of the model complexity [14]. And there will be a trade-off between the training loss, $\widehat{\mathcal{L}}_0^\gamma(Q)$, and the complexity (how far can the learned "posterior" $Q$ go from the "prior" $P$).

Uniquely for the non-IID semi-supervised setting, there is an extra term $D_{m,0}^\gamma(P;\lambda)$, which is the expected loss discrepancy between the target test subgroup $V_m$ and the training set $V_0$. Note that this quantity is independent of the training labels $y^0$. Not surprisingly, it is difficult to give generalization guarantees if the expected loss on $V_m$ is much larger than that on $V_0$ for any stochastic classifier $P$ independent of training data. We have to make some assumptions about the relationship between $V_m$ and $V_0$ to obtain a meaningful bound on $\frac{1}{\lambda}D_{m,0}^\gamma(P;\lambda)$, which we will discuss in details in Section 4.2.

**Deterministic classifier bound.** Utilizing standard techniques in PAC-Bayesian analysis [23, 27, 30], we can convert the bound for stochastic classifiers in Theorem 1 to a bound for deterministic classifiers as stated in Theorem 2 below (see Appendix A.2 for the proof).

**Theorem 2** (Subgroup Generalization of Deterministic Classifiers)**.** *Let $\tilde{h}$ be any classifier in $\mathcal{H}$. For any $0 < m \leq M$, for any $\lambda > 0$ and $\gamma \geq 0$, for any "prior" distribution $P$ on $\mathcal{H}$ that is independent of the training data on $V_0$, with probability at least $1 - \delta$ over the sample of $y^0$, for any $Q$ on $\mathcal{H}$ such that $\mathrm{Pr}_{h\sim Q}\left(\max_{i\in V_0\cup V_m}\|h_i(X,G) - \tilde{h}_i(X,G)\|_\infty < \frac{\gamma}{8}\right) > \frac{1}{2}$, we have*

$$\mathcal{L}_m^0(\tilde{h}) \leq \widehat{\mathcal{L}}_0^\gamma(\tilde{h}) + \frac{1}{\lambda}\left(2(D_{\mathrm{KL}}(Q\|P) + 1) + \ln\frac{1}{\delta} + \frac{\lambda^2}{4N_0} + D_{m,0}^{\gamma/2}(P;\lambda)\right). \tag{4}$$

Theorem 1 and 2 are not specific to GNNs and hold for any (respectively stochastic and deterministic) classifier under the semi-supervised setup. In Section 4.2, we will apply Theorem 2 to obtain a subgroup generalization bound that explicitly depends on the characteristics of GNNs and the data.

## 4.2 Subgroup Generalization Bound for Graph Neural Networks

**The GNN model.** We consider GNNs where the node feature aggregation step and the prediction step are separate. In particular, we assume the GNN classifier takes the form of $h_i(X,G) = f(g_i(X,G); W_1, W_2, \ldots, W_L)$, where $g$ is an aggregation function as we described in Section 3.1 and $f$ is a ReLU-activated $L$-layer Multi-Layer Perceptron (MLP) with $W_1, \ldots, W_L$ as parameters for each layer[8]. Denote the largest width of all the hidden layers as $b$.

**Remark 1.** *There is a technical restriction on the possible choice of the aggregation function $g$. For the following derivation of the generalization bound (6) to be valid, we need the condition that the node labels $y_i$'s are independent conditional on their aggregated features $g_i(X,G)$'s, as introduced in the problem formulation in Section 3.1. However, we also note that this condition tends to hold when the aggregated features $g_i(X,G)$'s contain rich information about the labels.*

**Upper-bounding $D_{m,0}^\gamma(P;\lambda)$.** To derive the generalization guarantee, we need to upper-bound the expected loss discrepancy $D_{m,0}^\gamma(P;\lambda)$. It turns out that we have to make some assumptions on the data in order to get a meaningful upper bound.

So far we have not had any restrictions on the conditional label distributions $\mathrm{Pr}(y_i = k \mid g_i(X,G))$. If the label distributions on $V \setminus V_0$ can be arbitrarily different from those on $V_0$, the generalization

---

[7]Theorem 1 also holds when we substitute $\mathcal{L}_m^{\gamma/2}(h)$ and $\mathcal{L}_m^{\gamma/2}(Q)$ as $\mathcal{L}_m^\gamma(h)$ and $\mathcal{L}_m^\gamma(Q)$ respectively. But we state Theorem 1 in this form to ease the presentation of the later analysis.

[8]SGC [45] and APPNP [21] are special cases of GNNs in this form.

can be arbitrarily poor. We therefore assume that the label distributions conditional on aggregated features are smooth (Assumption 1).

**Assumption 1** (Smoothness of Data Distribution). *Assume there exist $c$-Lipschitz continuous functions $\eta_1, \eta_2, \ldots, \eta_K : \mathbb{R}^{D'} \to [0, 1]$, such that, for any node $i \in V$,*

$$\Pr(y_i = k \mid g_i(X, G)) = \eta_k(g_i(X, G)), \forall k = 1, \ldots, K.$$

We also need to characterize the relationship between a target test subgroup $V_m$ and the training set $V_0$. For this purpose, we define the distance from $V_m$ to $V_0$ and the concept of *near set* below.

**Definition 2** (Distance To Training Set and Near Set). *For each $0 < m \leq M$, define the distance from the subgroup $V_m$ to the training set $V_0$ as*

$$\epsilon_m := \max_{j \in V_m} \min_{i \in V_0} \|g_i(X, G) - g_j(X, G)\|_2.$$

*Further, for each $i \in V_0$, define the near set of $i$ with respect to $V_m$ as*

$$V_m^{(i)} := \{j \in V_m \mid \|g_i(X, G) - g_j(X, G)\|_2 \leq \epsilon_m\}.$$

*Clearly,*

$$V_m = \cup_{i \in V_0} V_m^{(i)}.$$

Then, with the Assumption 2 and 3 below, we can bound the expected loss discrepancy $D_{m,0}^{\gamma}(P; \lambda)$ with the following Lemma 1 (see the proof in Appendix A.3).

**Assumption 2** (Equal-Sized and Disjoint Near Sets). *For any $0 < m \leq M$, assume the near sets of each $i \in V_0$ with respect to $V_m$ are disjoint and have the same size $s_m \in \mathbb{N}^+$.*

**Assumption 3** (Concentrated Expected Loss Difference). *Let $P$ be a distribution on $\mathcal{H}$, defined by sampling the vectorized MLP parameters from $\mathcal{N}(0, \sigma^2 I)$ for some $\sigma^2 \leq \frac{(\gamma/8\epsilon_m)^{2/L}}{2b(\lambda N_0^{-\alpha} + \ln 2bL)}$. For any $L$-layer GNN classifier $h \in \mathcal{H}$ with model parameters $W_1^h, \ldots, W_L^h$, define $T_h := \max_{l=1,\ldots,L} \|W_l\|_2$. Assume that there exists some $0 < \alpha < \frac{1}{4}$ satisfying*

$$\Pr_{h \sim P} \left( \mathcal{L}_m^{\gamma/4}(h) - \mathcal{L}_0^{\gamma/2}(h) > N_0^{-\alpha} + cK\epsilon_m \mid T_h^L \epsilon_m > \frac{\gamma}{8} \right) \leq e^{-N_0^{2\alpha}}.$$

**Lemma 1** (Bound for $D_{m,0}^{\gamma}(P; \lambda)$). *Under Assumption 1, 2 and 3, for any $0 < m \leq M$, any $0 < \lambda \leq N_0^{2\alpha}$ and $\gamma \geq 0$, assume the "prior" $P$ on $\mathcal{H}$ is defined by sampling the vectorized MLP parameters from $\mathcal{N}(0, \sigma^2 I)$ for some $\sigma^2 \leq \frac{(\gamma/8\epsilon_m)^{2/L}}{2b(\lambda N_0^{-\alpha} + \ln 2bL)}$. We have*

$$D_{m,0}^{\gamma/2}(P; \lambda) \leq \ln 3 + \lambda cK\epsilon_m. \tag{5}$$

Intuitively, what we need to bound $D_{m,0}^{\gamma}(P; \lambda)$ is that the training set $V_0$ is "representative" for $V_m$. This is reasonable in practice as it is natural to select the training samples according to the distribution of the population. Specifically, Assumption 2 assumes that $V_m$ can be split into equal-sized partitions indexed by the training samples. The elements of each partition $V_m^{(i)}$ are close to the corresponding training sample $i$ but not so close to training samples other than $i$. This assumption is stronger than needed to obtain a meaningful bound on $D_{m,0}^{\gamma}(P; \lambda)$, and we can relax it by only assuming that most samples in $V_m$ have proportional "close representatives" in $V_0$. But we keep Assumption 2 in this work, as it is intuitively clear and significantly eases the analysis and notations. Assumption 3 essentially assumes that the expected margin loss on $V_m$ is not much larger than that on $V_0$ when the number of samples becomes large. We first note that this assumption becomes trivially true in the degenerate case that all samples in $V_m$ and $V_0$ are IID. In this case, $\mathcal{L}_m^{\gamma/4}(h) = \mathcal{L}_0^{\gamma/4}(h) < \mathcal{L}_0^{\gamma/2}(h) \leq 0$ for any classifier $h$. In Appendix A.5, we further provide a simple non-IID example where Assumption 3 holds.

The bound (5) suggests that the closer between $V_m$ and $V_0$ (smaller $\epsilon_m$), the smaller the expected loss discrepancy.

**Bound for GNNs.** Finally, with an additional technical assumption (Assumption 4) that the maximum L2 norm of aggregated node features does not grow too fast in terms of the number of training samples, we obtain a subgroup generalization bound for GNNs in Theorem 3. The proof of Theorem 3 can be found in Appendix A.4.

**Assumption 4.** *Define $B_m := \max_{i \in V_0 \cup V_m} \|g_i(X, G)\|_2$. For any classifier $\tilde{h} \in \mathcal{H}$ with parameters $\{\widetilde{W}_l\}_{l=1}^L$, assume $\|\widetilde{W}_l\|_F \leq C$ for $l = 1, \dots, L$. Assume $B_m, C$ are constants with respect to $N_0$.*

**Theorem 3** (Subgroup Generalization Bound for GNNs). *Let $\tilde{h}$ be any classifier in $\mathcal{H}$ with parameters $\{\widetilde{W}_l\}_{l=1}^L$. Under Assumptions 1, 2, 3, and 4, for any $0 < m \leq M$, $\gamma \geq 0$, and large enough $N_0$, with probability at least $1 - \delta$ over the sample of $y^0$, we have*

$$\mathcal{L}_m^0(\tilde{h}) \leq \widehat{\mathcal{L}}_0^\gamma(\tilde{h}) + O\left( cK\epsilon_m + \frac{b\sum_{l=1}^L \|\widetilde{W}_l\|_F^2}{(\gamma/8)^{2/L} N_0^\alpha}(\epsilon_m)^{2/L} + \frac{1}{N_0^{1-2\alpha}} + \frac{1}{N_0^{2\alpha}} \ln \frac{LC(2B_m)^{1/L}}{\gamma^{1/L}\delta} \right).$$

(6)

Next, we investigate the qualitative implications of our theoretical results.

### 4.3 Implications for Fairness of Graph Neural Networks

**Theoretically predictable accuracy disparity.** One merit of our analysis is that we can apply Theorem 3 on different subgroups of the unlabeled nodes and compare the subgroup generalization bounds. This allows us to study the accuracy disparity across subgroups from a theoretical perspective.

A major factor that affects the generalization bound (6) is $\epsilon_m$, the aggregated-feature distance (in terms of $g(X, G)$) from the target test subgroup $V_m$ to the training set $V_0$. The generalization bound (6) suggests that there is a better generalization guarantee for subgroups that are closer to the training set. In other words, it is unfair for subgroups that are far away from the training set. While our theoretical analysis can only tell the difference among *upper bounds* of generalization errors, we empirically verify that, in the following Section 5, the aggregated-feature distance $\epsilon_m$ is indeed a strong predictor for the test accuracy of each subgroup $V_m$. More specifically, the test accuracy decreases as the distance increases, which is consistent with the theoretical prediction given by the bound (6).

**Impact of the structural positions of nodes.** We further investigate if the aggregated-feature distance can be related to simpler and more interpretable graph characteristics, in order to obtain a more intuitive understanding of how the structural positions of nodes influence the prediction accuracy on them. We find that the *geodesic distance* (length of the shortest path) between two nodes is positively related to the distance between their aggregated features in some scenarios[9], such as when the node features exhibit homophily [28]. Empirically, we also observe that test nodes with larger geodesic distance to the training set tend to suffer from lower accuracy (see Figure 2).

In contrast, we find that common node centrality metrics (e.g., degree and PageRank) have less influence on the test accuracy (see Figure 3). These centrality metrics only capture the graph characteristics of the test nodes alone, but do not take their relationship to the training set into account, which is a key factor suggested by our theoretical analysis.

**Impact of training data selection.** Another implication of the theoretical results is that the selection of the training set plays an important role on the fairness of the learned GNN models. First, if the training set is selected unevenly on the graph, leaving part of the test nodes far away, there will likely be a large accuracy disparity. Second, a key ingredient in the proof of Lemma 1 is that the GNN predictions on two nodes tend to be more similar if they are closer in terms of the aggregated node features. This suggests that, if an individual training node is close to many test nodes, it may bias the predictions of the learned GNN on the test nodes towards the class it belongs to.

## 5 Experiments

In this section, we empirically verify the fairness implications suggested by our theoretical analysis.

**General setup.** We experiment on 4 popular GNN models, GCN [20], GAT [41], SGC [45], and APPNP [21], as well as an MLP model for reference. For all models, we use the implementations by Deep Graph Library [43]. For each experiment setting, 40 independent trials are carried out.

### 5.1 Accuracy Disparity Across Subgroups

**Subgroups.** We examine the accuracy disparity with three types of subgroups as described below.

---

[9]A more detailed discussion on such scenarios is provided in Appendix D.1.

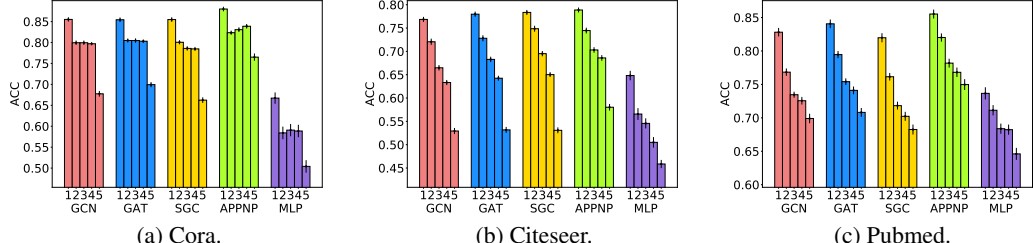

Figure 1: Test accuracy disparity across subgroups by aggregated-feature distance. Each figure corresponds to a dataset, and each bar cluster corresponds to a model. Bars labeled 1 to 5 represent subgroups with increasing distance to training set. Results are averaged over 40 independent trials with different random splits of the data, and the error bar represents the standard error of the mean.

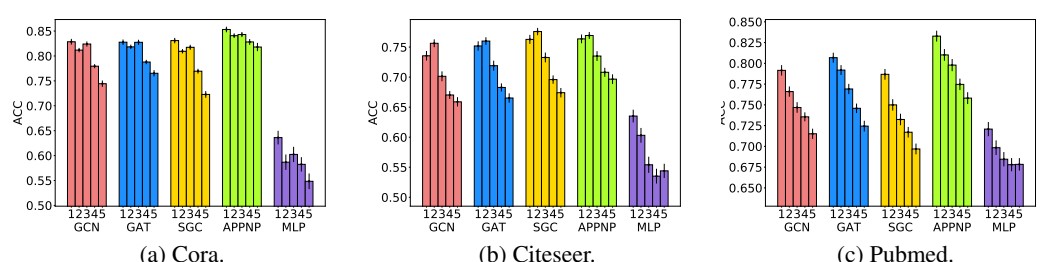

Figure 2: Test accuracy disparity across subgroups by geodesic distance. The experiment and plot settings are the same as Figure 1, except for the bars labeled from 1 to 5 here represent subgroups with increasing shortest-path distance to training set.

*Subgroup by aggregated-feature distance.* In order to directly investigate the effect of $\epsilon_m$ on the generalization bound (6), we first split the test nodes into subgroups by their distance to the training set in terms of the aggregated features. We use the two-step aggregated features to calculate the distance. In particular, denote the adjacency matrix of the graph $G$ as $A \in \{0, 1\}^{N \times N}$ and the corresponding degree matrix as $D$, where $D$ is an $N \times N$ diagonal matrix with $D_{ii} = \sum_{j=1}^{N} A_{ij}, \forall i = 1, \ldots, N$. Given the feature matrix $X \in \mathbb{R}^{N \times D}$, the two-step aggregated features $Z$ are obtained by $Z = (D + I)^{-1}(A + I)(D + I)^{-1}(A + I)X$. For each test node $i$, we calculate its aggregated-feature distance to the training set $V_0$ as $d_i = \min_{j \in V_0} \|Z_i - Z_j\|_2$. Then we sort the test nodes according to this distance and split them into 5 equal-sized subgroups.

Strictly speaking, our theory does not directly apply to GCN and GAT as they are not in the form as we defined in Section 4.2. Moreover, the two-step aggregated feature does not match exactly to the feature aggregation function of SGC and APPNP. Nevertheless, we find that even with such approximations, we are still able to observe the expected descending trend of test accuracy with respect to increasing distance in terms of the two-step aggregated features, on all four GNN models.

*Subgroup by geodesic distance.* As we discussed in Section 4.3, geodesic distance on the graph may well relate to the aggregated-feature distance. So we also define subgroups based on geodesic distance. We split the subgroups by replacing the aggregated-feature distance $d_i$ of each test node $i$ with the minimum of the geodesic distances from $i$ to each training node on the graph.

*Subgroup by node centrality.* Lastly, we define subgroups based on 4 types of common node centrality metrics (degree, closeness, betweenness, and PageRank) of the test nodes. We split the subgroups by replacing the aggregated-feature distance $d_i$ of each test node $i$ with the centrality score of $i$. The purpose of this setup is to show that the common node centrality metrics are not sufficient to capture the monotonic trend of test accuracy.

**Experiment setup.** Following common GNN experiment setup [38], we randomly select 20 nodes in each class for training, 500 nodes for validation, and 1,000 nodes for testing. Once training is done,

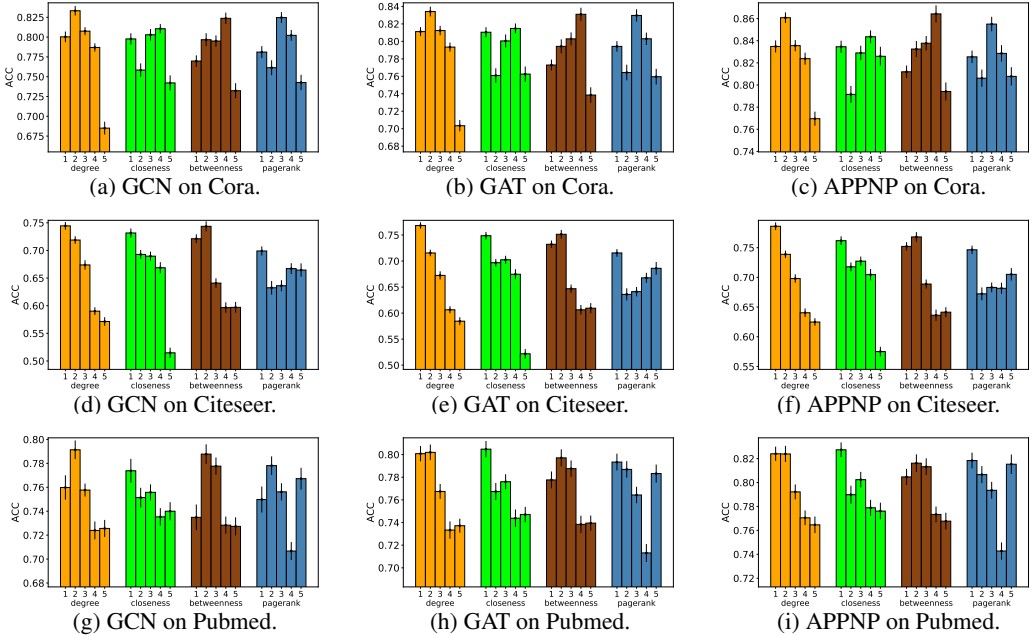

Figure 3: Test accuracy disparity across subgroups by node centrality. Each figure corresponds to the results of a pair of model and dataset, and each bar cluster corresponds to the subgroups defined by a certain centrality metric. In each cluster, the bars labeled from 1 to 5 represent subgroups with decreasing node centrality. Other settings are the same as Figure 1.

we report the test accuracy on subgroups defined by aggregated-feature distance, geodesic distance, and node centrality in Figure 1, 2, and 3 respectively[10].

**Experiment results.** First, as shown in Figure 1, there is a clear trend that the accuracy of a test subgroup decreases as the aggregated-feature distance between the test subgroup and the training set increases. And the trend is consistent for all 4 GNN models on all the datasets we test on (except for APPNP on Cora). This result verifies the existence of accuracy disparity suggested by Theorem 3.

Second, we observe in Figure 2 that there is a similar trend when we split subgroups by the geodesic distance. This suggests that the geodesic distance on the graph can sometimes be used as a simpler indicator in practice for machine learning fairness on graph-structured data. Using such a classical graph metric as an indicator also helps us connect graph-based machine learning to network theory, especially to understandings about social networks, to better analyze fairness issues of machine learning on social networks, where high-stake decisions related to human subjects may be involved.

Furthermore, as shown in Figure 3, there is no clear monotonic trend for test accuracy when we split subgroups by node centrality, except for some particular combinations of GNN model and dataset. Empirically, the common node centrality metrics are not as good as the geodesic distance in terms of capturing the accuracy disparity. This contrast highlights the importance of the insight provided by our analysis: the "distance" to the training set, rather than some graph characteristics of the test nodes alone, is the key predictor of test accuracy.

Finally, it is intriguing that, in both Figure 1 and 2, the test accuracy of MLP (which does not use the graph structure) also decreases as the distance of a subgroup to the training set increases. This result is perhaps not surprising if the subgroups were defined by distance on the original node features, as MLP can be viewed as a special GNN where the feature aggregation function is an identity mapping, so the "aggregated features" for MLP essentially equal to the original features. Our theoretical analysis can then be similarly applied to MLP. The question is why there is also an accuracy disparity w.r.t. the aggregated-feature distance and the geodesic distance. We suspect this is because these datasets present homophily, i.e., original (non-aggregated) features of geodesically closer nodes tend to be more similar. As a result, a subgroup with smaller geodesic distance may also have closer node

---

[10]The main paper reports the results on a small set of datasets (Cora, Citeseer, and Pubmed [37, 47]). Results on more datasets, including large-scale datasets from Open Graph Benchmark [17], are shown in Appendix C.

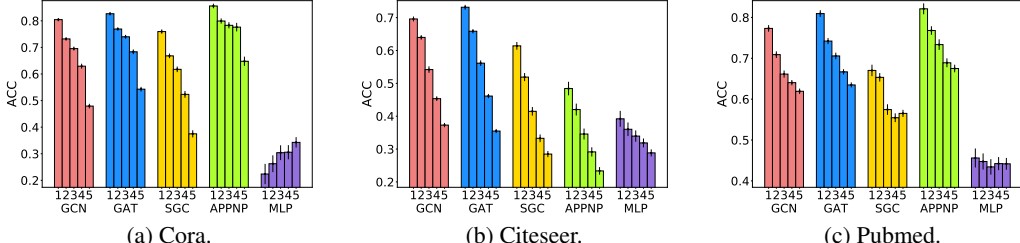

| (a) Cora. | (b) Citeseer. | (c) Pubmed. |

Figure 4: Test accuracy disparity across subgroups by aggregated-feature distance, experimented with noisy features. The experiment and plot settings are the same as Figure 1, except for the node features are perturbed by independent noises such that they are less homophilious.

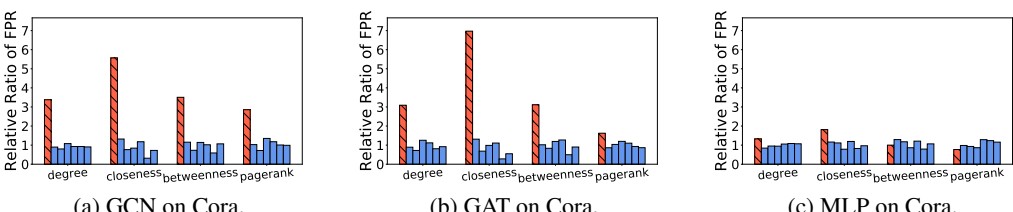

| (a) GCN on Cora. | (b) GAT on Cora. | (c) MLP on Cora. |

Figure 5: Relative ratio between the FPR under biased training node selection and the FPR under uniform training node selection. Each bar in each cluster corresponds to a class (there are 7 classes in total). The red shaded bar indicates the class with high centrality training nodes under the biased setup. Each cluster corresponds to a centrality metric being used for the biased node selection.

features to the training set. To verify this hypothesis, we repeat the experiments in Figure 1, but with independent noises added to node features such that they become less homophilious. As in Figure 4, the decreasing pattern of test accuracy across subgroups remains for the 4 GNNs on all datasets; while for MLP, the pattern disappears on Cora and Pubmed and becomes less sharp on Citeseer.

## 5.2 Impact of Biased Training Node Selection

In all the previous experiments, we follow the standard GNN training setup where 20 training nodes are uniformly sampled for each class. Next we investigate the impact if the selection of training nodes is biased, verifying our discussions in Section 4.3. We will demonstrate that the node centrality scores of the training nodes play an important role in the learned GNN model.

We choose a "dominant class" and construct a manipulated training set. For each class, we still sample 20 training nodes but in a biased way. For the dominant class, the sample is biased towards nodes of high centrality; while for other classes, the sample is biased towards nodes of low centrality. We evaluate the relative ratio of False Positive Rate (FPR) for each class between the setup using manipulated training set and the setup using uniformly sampled training set.

As shown in Figure 5, compared to MLP, the GNN models have significantly worse FPR for the dominant class when the training nodes are biased. This is because, after feature aggregation, there will be a larger proportion of test nodes that are closer to the training nodes of higher centrality. And the learned GNN model will be heavily biased towards the training labels of these nodes.

## 6 Conclusion

We present a novel PAC-Bayesian analysis for the generalization ability of GNNs on node-level semi-supervised learning tasks. As far as we know, this is the first generalization bound for GNNs for non-IID node-level tasks without strong assumptions on the data generating mechanism. One advantage of our analysis is that it can be applied to arbitrary subgroups of the test nodes, which allows us to investigate an accuracy-disparity style of fairness for GNNs. Both the theoretical and empirical results suggest that there is an accuracy disparity across subgroups of test nodes that have varying distance to the training set, and nodes with larger distance to the training nodes suffer from a lower classification accuracy. In the future, we would like to utilize our theory to analyze the fairness of GNNs on real-world applications and develop principled methods to mitigate the unfairness.

**Acknowledgements**

This work was in part supported by the National Science Foundation under grant number 1633370. The authors claim no competing interests.

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
