# Contents

# A Proofs

## A.1 Proof of Theorem 1

We first introduce three lemmas whose proofs can be found in the referred literature.

**Lemma 2** (Hoeffding's Inequality for Bounded Random Variables [15]). *Suppose $X_1, X_2, \ldots, X_n$ are independent random variables with $a_i \leq X_i \leq b_i, \forall i = 1, 2 \ldots, n$. Let $\bar{X} = \frac{1}{n} \sum_{i=1}^{n} X_i$. Then, for any $t > 0$,*

$$\Pr\big(|\bar{X} - \mathbb{E}\bar{X}| > t\big) \leq 2e^{-\frac{n^2 t^2}{\Sigma_{i=1}^{n}(b_i - a_i)^2}}.$$

**Lemma 3** (Sub-Gaussianity). *If $X$ is a centered random variable, i.e., $\mathbb{E}X = 0$, and if $\exists \nu > 0$, for any $t > 0$,*

$$\Pr(|X| > t) \leq 2e^{-\nu t^2}.$$

*Then, for any $\lambda > 0$,*

$$\mathbb{E}e^{\lambda X} \leq e^{\frac{\lambda^2}{2\nu}}.$$

See Rivasplata [33] (Theorem 3.1) for the proof of Lemma 3.

**Lemma 4** (Change-of-Measure Inequality, Lemma 17 in Germain et al. [12]). *For any two distributions $P$ and $Q$ defined on $\mathcal{H}$, and any function $\psi : \mathcal{H} \to \mathbb{R}$,*

$$\mathbb{E}_{h \sim Q}[\psi(h)] \leq D_{\mathrm{KL}}(Q\|P) + \ln \mathbb{E}_{h \sim P}[e^{\psi(h)}].$$

Then we can prove Theorem 1. For convenience, we re-state it as Theorem 4 below.

**Theorem 4** (Subgroup Generalization of Stochastic Classifiers). *For any $0 < m \leq M$, for any $\lambda > 0$ and $\gamma \geq 0$, for any "prior" distribution $P$ on $\mathcal{H}$ that is independent of the training data on $V_0$, with probability at least $1 - \delta$ over the sample of $y^0$, for any $Q$ on $\mathcal{H}$, we have[11]*

$$\mathcal{L}_m^{\gamma/2}(Q) \leq \widehat{\mathcal{L}}_0^{\gamma}(Q) + \frac{1}{\lambda} \left( D_{\mathrm{KL}}(Q\|P) + \ln \frac{1}{\delta} + \frac{\lambda^2}{4N_0} + D_{m,0}^{\gamma}(P; \lambda) \right). \tag{7}$$

*Proof.* We prove the result by upper-bounding the quantity $\lambda(\mathcal{L}_m^{\gamma/2}(Q) - \widehat{\mathcal{L}}_0^{\gamma}(Q))$. First, we have

$$\lambda(\mathcal{L}_m^{\gamma/2}(Q) - \widehat{\mathcal{L}}_0^{\gamma}(Q))$$
$$\leq \mathbb{E}_{h \sim Q} \lambda(\mathcal{L}_m^{\gamma/2}(h) - \widehat{\mathcal{L}}_0^{\gamma}(h))$$
$$\leq D_{\mathrm{KL}}(Q\|P) + \ln \mathbb{E}_{h \sim P} e^{\lambda\left(\mathcal{L}_m^{\gamma/2}(h) - \widehat{\mathcal{L}}_0^{\gamma}(h)\right)}, \tag{8}$$

where the first inequality is due to Jensen's inequality, and the last inequality is due to Lemma 4.

Next we would like to upper-bound the second term in the RHS of (8). Note that the quantity $U := \mathbb{E}_{h \sim P} e^{\lambda\left(\mathcal{L}_m^{\gamma/2}(h) - \widehat{\mathcal{L}}_0^{\gamma}(h)\right)}$ is a random variable with the randomness coming from the sample of node labels $y^0$ for $V_0$. Also note that $P$ is independent of $y^0$. Applying Markov's inequality to $U$, we have for any $\delta > 0$, with probability at least $1 - \delta$ over the sample of $y^0$,

$$U \leq \frac{1}{\delta} \mathbb{E}_{y^0} U,$$

and hence,

$$\ln U \leq \ln \frac{1}{\delta} \mathbb{E}_{y^0} U = \ln \frac{1}{\delta} + \ln \mathbb{E}_{y^0} U.$$

Then we need to upper-bound the quantity $\ln \mathbb{E}_{y^0} U$. We can re-write it as

$$\ln \mathbb{E}_{y^0} U = \ln \mathbb{E}_{y^0} \mathbb{E}_{h \sim P} e^{\lambda\left(\mathcal{L}_m^{\gamma/2}(h) - \widehat{\mathcal{L}}_0^{\gamma}(h)\right)} = \ln \mathbb{E}_{h \sim P} \mathbb{E}_{y^0} e^{\lambda\left(\mathcal{L}_m^{\gamma/2}(h) - \widehat{\mathcal{L}}_0^{\gamma}(h)\right)}. \tag{9}$$

---

[11]Theorem 4 also holds when we substitute $\mathcal{L}_m^{\gamma/2}(h)$ and $\mathcal{L}_m^{\gamma/2}(Q)$ as $\mathcal{L}_m^{\gamma}(h)$ and $\mathcal{L}_m^{\gamma}(Q)$ respectively. But we state the theorem in this form to ease the development of the later analysis.

For a fixed model $h$,

$$\mathbb{E}_{y^0} e^{\lambda\left(\mathcal{L}_m^{\gamma/2}(h) - \widehat{\mathcal{L}}_0^{\gamma}(h)\right)}$$

$$= \mathbb{E}_{y^0} e^{\lambda\left(\mathcal{L}_m^{\gamma/2}(h) - \mathcal{L}_0^{\gamma}(h) + \mathcal{L}_0^{\gamma}(h) - \widehat{\mathcal{L}}_0^{\gamma}(h)\right)}$$

$$= \mathbb{E}_{y^0} e^{\lambda\left(\mathcal{L}_m^{\gamma/2}(h) - \mathcal{L}_0^{\gamma}(h)\right)} e^{\lambda\left(\mathcal{L}_0^{\gamma}(h) - \widehat{\mathcal{L}}_0^{\gamma}(h)\right)}$$

$$= e^{\lambda\left(\mathcal{L}_m^{\gamma/2}(h) - \mathcal{L}_0^{\gamma}(h)\right)} \mathbb{E}_{y^0} e^{\lambda\left(\mathcal{L}_0^{\gamma}(h) - \widehat{\mathcal{L}}_0^{\gamma}(h)\right)}. \tag{10}$$

In the following we will give an upper bound on $\mathbb{E}_{y^0} e^{\lambda\left(\mathcal{L}_0^{\gamma}(h) - \widehat{\mathcal{L}}_0^{\gamma}(h)\right)}$ that is independent of $h$. Recall that

$$\widehat{\mathcal{L}}_0^{\gamma}(h) = \frac{1}{N_0} \sum_{i \in V_0} \mathbb{1}\left[h_i(X, G)[y_i] \leq \gamma + \max_{k \neq y_i} h_i(X, G)[k]\right],$$

where the node labels are independently sampled (though not from the identical distribution), so $\widehat{\mathcal{L}}_0^{\gamma}(h)$ is the empirical mean of $N_0$ independent Bernoulli random variables and $\mathcal{L}_0^{\gamma}(h)$ is the expectation of $\widehat{\mathcal{L}}_0^{\gamma}(h)$. By Lemma 2, for any $t > 0$,

$$\Pr\left(|\mathcal{L}_0^{\gamma}(h) - \widehat{\mathcal{L}}_0^{\gamma}(h)| \geq t\right) \leq 2e^{-2N_0 t^2},$$

and hence $\mathcal{L}_0^{\gamma}(h) - \widehat{\mathcal{L}}_0^{\gamma}(h)$ is sub-Gaussian. Further by Lemma 3, we have

$$\mathbb{E}_{y^0} e^{\lambda\left(\mathcal{L}_0^{\gamma}(h) - \widehat{\mathcal{L}}_0^{\gamma}(h)\right)} \leq e^{\frac{\lambda^2}{4N_0}},$$

which implies that

$$\mathbb{E}_{y^0} e^{\lambda\left(\mathcal{L}_0^{\gamma}(h) - \widehat{\mathcal{L}}_0^{\gamma}(h)\right)} \leq e^{\frac{\lambda^2}{4N_0}}, \tag{11}$$

Therefore, plugging (10) and (11) back into (9), we have

$$\ln \mathbb{E}_{y^0} U$$

$$\leq \ln \mathbb{E}_{h \sim P} e^{\lambda\left(\mathcal{L}_m^{\gamma/2}(h) - \mathcal{L}_0^{\gamma}(h)\right)} e^{\frac{\lambda^2}{4N_0}}$$

$$= D_{m,0}^{\gamma}(P; \lambda) + \frac{\lambda^2}{4N_0}.$$

Finally, plugging everything back into (8), we get

$$\lambda(\mathcal{L}_m^{\gamma/2}(Q) - \widehat{\mathcal{L}}_0^{\gamma}(Q))$$

$$\leq D_{\mathrm{KL}}(Q\|P) + \ln \mathbb{E}_{h \sim P} e^{\lambda\left(\mathcal{L}_m^{\gamma/2}(h) - \widehat{\mathcal{L}}_0^{\gamma}(h)\right)}$$

$$\leq D_{\mathrm{KL}}(Q\|P) + \ln\frac{1}{\delta} + \frac{\lambda^2}{4N_0} + D_{m,0}^{\gamma}(P; \lambda).$$

Rearranging the terms gives us the final result

$$\mathcal{L}_m^{\gamma/2}(Q) \leq \widehat{\mathcal{L}}_0^{\gamma}(Q) + \frac{1}{\lambda}\left(D_{\mathrm{KL}}(Q\|P) + \ln\frac{1}{\delta} + \frac{\lambda^2}{4N_0} + D_{m,0}^{\gamma}(P; \lambda)\right).$$

$\square$

## A.2 Proof of Theorem 2

We re-state Theorem 2 as Theorem 5 below.

**Theorem 5** (Subgroup Generalization of Deterministic Classifiers). *Let $\tilde{h}$ be any classifier in $\mathcal{H}$. For any $0 < m \leq M$, for any $\lambda > 0$ and $\gamma \geq 0$, for any "prior" distribution $P$ on $\mathcal{H}$ that is independent of the training data on $V_0$, with probability at least $1 - \delta$ over the sample of $y^0$, for any $Q$ on $\mathcal{H}$ such that $\Pr_{h \sim Q}\left(\max_{i \in V_0 \cup V_m} \|h_i(X, G) - \tilde{h}_i(X, G)\|_\infty < \frac{\gamma}{8}\right) > \frac{1}{2}$, we have*

$$\mathcal{L}_m^0(\tilde{h}) \leq \widehat{\mathcal{L}}_0^{\gamma}(\tilde{h}) + \frac{1}{\lambda}\left(2(D_{\mathrm{KL}}(Q\|P) + 1) + \ln\frac{1}{\delta} + \frac{\lambda^2}{4N_0} + D_{m,0}^{\gamma/2}(P; \lambda)\right). \tag{12}$$

*Proof.* For simplicity, we write $h_i(X, G)$ and $\tilde{h}_i(X, G)$ as $h_i$ and $\tilde{h}_i$ in this proof. We first construct a distribution $Q'$ by restricting $Q$ on $\mathcal{H}_{\tilde{h}} \subseteq \mathcal{H}$, where

$$\mathcal{H}_{\tilde{h}} := \{h \in \mathcal{H} \mid \max_{i \in V_0 \cup V_m} \|h_i - \tilde{h}_i\|_\infty < \frac{\gamma}{8}\}.$$

And $Q'$ is defined as

$$Q'(h) = \begin{cases} \frac{1}{Z_{Q'}} Q(h), & \text{if } h \in \mathcal{H}_{\tilde{h}} \\ 0, & \text{otherwise} \end{cases},$$

where $Z_{Q'} = \Pr_{h \sim Q}(h \in \mathcal{H}_{\tilde{h}}) \geq \frac{1}{2}$ by the condition of the theorem.

For any $h \in \mathcal{H}_{\tilde{h}}$ and any sample $i \in V_0 \cup V_m$, by definition of $\mathcal{H}_{\tilde{h}}$, we have

$$\max_{k,k' \in \{1,\dots,K\}} |(\tilde{h}_i[k] - \tilde{h}_i[k']) - (h_i[k] - h_i[k'])| < \frac{\gamma}{4},$$

which implies the following relationships:

$$\mathcal{L}_m^0(\tilde{h}) \leq \mathcal{L}_m^{\gamma/4}(h), \quad \widehat{\mathcal{L}}_0^{\gamma/2}(h) \leq \widehat{\mathcal{L}}_0^\gamma(\tilde{h}).$$

Therefore, with probability at least $1 - \delta$ over the sample of $y^m$,

$$\mathcal{L}_m^0(\tilde{h})$$
$$\leq \mathbb{E}_{h \sim Q'} \mathcal{L}_m^{\gamma/4}(h)$$
$$\leq \mathbb{E}_{h \sim Q'} \widehat{\mathcal{L}}_0^{\gamma/2}(h) + \frac{1}{\lambda} \left( D_{\mathrm{KL}}(Q'\|P) + \ln \frac{1}{\delta} + \frac{\lambda^2}{4N_0} + D_{m,0}^{\gamma/2}(P; \lambda) \right)$$
$$\leq \widehat{\mathcal{L}}_0^\gamma(\tilde{h}) + \frac{1}{\lambda} \left( D_{\mathrm{KL}}(Q'\|P) + \ln \frac{1}{\delta} + \frac{\lambda^2}{4N_0} + D_{m,0}^{\gamma/2}(P; \lambda) \right),$$

where the second inequality is due to the application of Theorem 1 by substituting $\gamma$ as $\gamma/2$ and $Q$ as $Q'$.

Finally, to complete the proof, we only need to show

$$D_{\mathrm{KL}}(Q'\|P) \leq 2(D_{\mathrm{KL}}(Q\|P) + 1).$$

Denote $\mathcal{H}_{\tilde{h}}^c$ as the complement of $\mathcal{H}_{\tilde{h}}$ and define $Q'^c$ as the distribution restricted to $\mathcal{H}_{\tilde{h}}^c$ similarly as $Q'$. Define $H(x) := -x \ln x - (1 - x) \ln(1 - x)$, which is the binary entropy function and we know $H(Z) \leq 1$. Then

$$D_{\mathrm{KL}}(Q\|P) = \int_{\mathcal{H}_{\tilde{h}}} \ln \frac{dQ}{dP} dQ + \int_{\mathcal{H}_{\tilde{h}}^c} \ln \frac{dQ}{dP} dQ$$
$$= Z_{Q'} \int_{\mathcal{H}} \ln \frac{dQ'}{dP} dQ' + (1 - Z_Q') \int_{\mathcal{H}} \ln \frac{dQ'^c}{dP} dQ'^c - H(Z_{Q'})$$
$$= Z_{Q'} D_{\mathrm{KL}}(Q'\|P) + (1 - Z_Q') D_{\mathrm{KL}}(Q'^c\|P) - H(Z_{Q'}).$$

So

$$D_{\mathrm{KL}}(Q'\|P) = \frac{1}{Z_{Q'}} \left( D_{\mathrm{KL}}(Q\|P) + H(Z_{Q'}) - (1 - Z_Q') D_{\mathrm{KL}}(Q'^c\|P) \right) \leq 2(D_{\mathrm{KL}}(Q\|P) + 1),$$

since $D_{\mathrm{KL}}(Q'^c\|P) \geq 0$. $\qquad\square$

## A.3 Proof of Lemma 1

We first present the following Lemma 5 that bounds the difference between the margin loss on $V_m$ and that on $V_0$ for a fixed GNN.

**Lemma 5.** *Suppose an $L$-layer GNN classifier $h$ is associated with model parameters $W_1, \dots, W_L$. Define $T_h := \max_{l=1,\dots,L} \|W_l\|_2$. Under Assumption 1 and 2, for any $0 < m \leq M$ and $\gamma \geq 0$, if $\epsilon_m T_h^L \leq \frac{\gamma}{4}$, then*

$$\mathcal{L}_m^{\gamma/2}(h) - \mathcal{L}_0^\gamma(h) \leq cK\epsilon_m.$$

*Proof.* For simplicity in this proof, for any $i \in V_0 \cup V_m$ and $k = 1, \ldots, K$, we use $h_i$ to denote $h_i(X, G)$ and use $\eta_k(i)$ to denote $\Pr(y_i = k \mid g_i(X, G))$. And define $\mathcal{L}^\gamma(h_i, y_i) := \mathbb{1}\left[h_i[y_i] \le \gamma + \max_{k \ne y_i} h_i[k]\right]$. Then we can write

$$\mathcal{L}_m^{\gamma/2}(h) - \mathcal{L}_0^\gamma(h)$$

$$=\mathbb{E}_{y^m}\left[\frac{1}{N_m}\sum_{j \in V_m}\mathcal{L}^{\gamma/2}(h_j, y_j)\right] - \mathbb{E}_{y^0}\left[\frac{1}{N_0}\sum_{i \in V_0}\mathcal{L}^\gamma(h_i, y_i)\right]$$

$$=\frac{1}{N_0}\mathbb{E}_{y^0, y^m}\sum_{i \in V_0}\frac{1}{s_m}\left(\sum_{j \in V_m^{(i)}}\mathcal{L}^{\gamma/2}(h_j, y_j)\right) - \mathcal{L}^\gamma(h_i, y_i)$$

where in the last step we have used Assumption 2. Therefore,

$$\mathcal{L}_m^{\gamma/2}(h) - \mathcal{L}_0^\gamma(h)$$

$$=\frac{1}{N_0}\sum_{i \in V_0}\frac{1}{s_m}\left(\sum_{j \in V_m^{(i)}}\mathbb{E}_{y_j}\mathcal{L}^{\gamma/2}(h_j, y_j)\right) - \mathbb{E}_{y_i}\mathcal{L}^\gamma(h_i, y_i)$$

$$=\frac{1}{N_0}\sum_{i \in V_0}\frac{1}{s_m}\left(\sum_{j \in V_m^{(i)}}\sum_{k=1}^{K}\eta_k(j)\mathcal{L}^{\gamma/2}(h_j, k)\right) - \sum_{k=1}^{K}\Pr(y_i = k)\mathcal{L}^\gamma(h_i, k)$$

$$=\frac{1}{N_0}\sum_{i \in V_0}\frac{1}{s_m}\sum_{j \in V_m^{(i)}}\sum_{k=1}^{K}\left(\eta_k(j)\mathcal{L}^{\gamma/2}(h_j, k) - \eta_k(i)\mathcal{L}^\gamma(h_i, k)\right)$$

$$=\frac{1}{N_0}\sum_{i \in V_0}\frac{1}{s_m}\sum_{j \in V_m^{(i)}}\sum_{k=1}^{K}\left(\eta_k(j)\left(\mathcal{L}^{\gamma/2}(h_j, k) - \mathcal{L}^\gamma(h_i, k)\right) + (\eta_k(j) - \eta_k(i))\mathcal{L}^\gamma(h_i, k)\right) \quad (13)$$

$$\le\frac{1}{N_0}\sum_{i \in V_0}\frac{1}{s_m}\sum_{j \in V_m^{(i)}}\sum_{k=1}^{K}\left(1 \cdot \left(\mathcal{L}^{\gamma/2}(h_j, k) - \mathcal{L}^\gamma(h_i, k)\right) + (\eta_k(j) - \eta_k(i)) \cdot 1\right), \quad (14)$$

where the last inequality utilizes the facts that both $\eta_k(j)$ and $\mathcal{L}^\gamma(h_i, k)$ are upper-bounded by 1. According to Assumption 1 and 2,

$$\eta_k(j) - \eta_k(i) \le c\|g_j(X, G) - g_i(X, G)\|_2 \le c\epsilon_m.$$

Further, as $h_i = f(g_i(X, G); W_1, \ldots, W_L)$ where $f$ is a ReLU-activated MLP, so

$$\|h_i - h_j\|_\infty \le \|g_i(X, G) - g_j(X, G)\|_2 \prod_{l=1}^{L}\|W_l\|_2 \le \epsilon_m T_h^L \le \frac{\gamma}{4}.$$

This implies that, for any $k = 1, \ldots, K$,

$$\mathcal{L}^{\gamma/2}(h_j, k) \le \mathcal{L}^\gamma(h_i, k).$$

So we have

$$\mathcal{L}_m^{\gamma/2}(h) - \mathcal{L}_0^\gamma(h)$$

$$\le\frac{1}{N_0}\sum_{i \in V_0}\frac{1}{s_m}\sum_{j \in V_m^{(i)}}\sum_{k=1}^{K}0 + c\epsilon_m$$

$$=cK\epsilon_m.$$

$\square$

Then we can prove Lemma 1, which we re-state as Lemma 6 below.

**Lemma 6** (Bound for $D_{m,0}^\gamma(P;\lambda)$). *Under Assumption 1, 2 and 3, for any $0 < m \le M$, any $0 < \lambda \le N_0^{2\alpha}$ and $\gamma \ge 0$, assume the "prior" $P$ on $\mathcal{H}$ is defined by sampling the vectorized MLP parameters from $\mathcal{N}(0, \sigma^2 I)$ for some $\sigma^2 \le \frac{(\gamma/8\epsilon_m)^{2/L}}{2b(\lambda N_0^{-\alpha} + \ln 2bL)}$. We have*

$$D_{m,0}^{\gamma/2}(P;\lambda) \le \ln 3 + \lambda cK\epsilon_m. \tag{15}$$

*Proof.* Recall that $D_{m,0}^{\gamma/2}(P;\lambda) = \ln \mathbb{E}_{h\sim P} e^{\lambda\left(\mathcal{L}_m^{\gamma/4}(h) - \mathcal{L}_0^{\gamma/2}(h)\right)}$. We prove the upper bound of $D_{m,0}^{\gamma/2}(P;\lambda)$ by decomposing the space $\mathcal{H}$ into the two regimes: a regime with bounded spectral norms of the model parameters required by Lemma 5, and its complement. Following Lemma 5, for any classifier $h$ with parameters $W_1, \dots, W_L$, we define $T_h := \max_{l=1,\dots,L} \|W_l\|_2$.

We first prove an upper bound on the probability $\Pr\left(T_h^L \epsilon_m > \frac{\gamma}{8}\right)$ over the drawing of $h \sim P$. For any $h$, as its vectorized MLP parameters $\mathrm{vec}(W_l)$, for each $l = 1, \dots, L$, is sampled from $\mathcal{N}(0, \sigma^2 I)$, we have the following spectral norm bound [40], for any $t > 0$,

$$\Pr(\|W_l\|_2 > t) \le 2be^{-\frac{t^2}{2b\sigma^2}},$$

where $b$ is the maximum width of all hidden layers of the MLP. Setting $t = \left(\frac{\gamma}{8\epsilon_m}\right)^{1/L}$ and applying a union bound, we have that

$$\Pr\left(T_h^L \epsilon_m > \frac{\gamma}{8}\right) = \Pr\left(T_h > \left(\frac{\gamma}{8\epsilon_m}\right)^{1/L}\right) \le 2bLe^{-\frac{(\gamma/8\epsilon_m)^{2/L}}{2b\sigma^2}} \le e^{-\lambda N_0^{-\alpha}},$$

where the last inequality utilizes the condition $\sigma^2 \le \frac{(\gamma/8\epsilon_m)^{2/L}}{2b(\lambda N_0^{-\alpha} + \ln 2bL)}$.

For any $h$ satisfying $T_h^L \epsilon_m \le \frac{\gamma}{8}$, by Lemma 5, we know that $e^{\lambda\left(\mathcal{L}_m^{\gamma/4}(h) - \mathcal{L}_0^{\gamma/2}(h)\right)} \le e^{\lambda cK\epsilon_m}$. For all $h$ such that $T_h^L \epsilon_m > \frac{\gamma}{8}$, by Assumption 3, with probability at least $1 - e^{-N_0^{2\alpha}}$,

$$e^{\lambda\left(\mathcal{L}_m^{\gamma/4}(h) - \mathcal{L}_0^{\gamma/2}(h)\right)} \le e^{\lambda N_0^{-\alpha} + \lambda cK\epsilon_m}.$$

Also note that $\mathcal{L}_m^{\gamma/4}(h) - \mathcal{L}_0^{\gamma/2}(h) \le 1$ trivially holds for any $h$. Therefore we have

$$D_{m,0}^{\gamma/2}(P;\lambda)$$
$$= \ln \mathbb{E}_{h\sim P} e^{\lambda\left(\mathcal{L}_m^{\gamma/4}(h) - \mathcal{L}_0^{\gamma/2}(h)\right)}$$
$$\le \ln\left(\Pr\left(T_h^L \epsilon_m > \frac{\gamma}{8}\right)\left(e^{-N_0^{2\alpha}} \cdot e^\lambda + (1 - e^{-N_0^{2\alpha}}) \cdot e^{\lambda N_0^{-\alpha} + \lambda cK\epsilon_m}\right) + \Pr\left(T_h^L \epsilon_m \le \frac{\gamma}{8}\right)e^{\lambda cK\epsilon_m}\right)$$
$$\le \ln\left(e^{\lambda - N_0^{2\alpha}} + \Pr\left(T_h^L \epsilon_m > \frac{\gamma}{8}\right)e^{\lambda N_0^{-\alpha}} e^{\lambda cK\epsilon_m} + e^{\lambda cK\epsilon_m}\right)$$
$$\le \ln\left(1 + e^{-\lambda N_0^{-\alpha}} e^{\lambda N_0^{-\alpha}} e^{\lambda cK\epsilon_m} + e^{\lambda cK\epsilon_m}\right)$$
$$= \ln\left(1 + 2e^{\lambda cK\epsilon_m}\right)$$
$$\le \ln 3 + \lambda cK\epsilon_m,$$
since $1 + 2e^{\lambda cK\epsilon_m} \le 3e^{\lambda cK\epsilon_m}$.

$\square$

## A.4  Proof of Theorem 3

The proof of Theorem 3 relies on the combination of Theorem 2, Lemma 1, and an intermediate result of the Theorem 1 in Neyshabur et al. [30] (which we state as Lemma 7 below).

**Lemma 7** (Neyshabur et al. [30]). *Let $\tilde{h}$ be any classifier in $\mathcal{H}$ with parameters $\{\widetilde{W}_l\}_{l=1}^L$. Define $\tilde{\beta} = \left(\prod_{l=1}^L \|\widetilde{W}_l\|_2\right)^{1/L}$. Let $\{U_l\}_{l=1}^L$ be the random perturbation to be added to $\{\widetilde{W}_l\}_{l=1}^L$ and $\mathrm{vec}(\{U_l\}_{l=1}^L) \sim \mathcal{N}(0, \sigma^2 I)$. Define $B_m := \max_{i \in V_0 \cup V_m} \|g_i(X, G)\|_2$. If*

$$\sigma \le \frac{\gamma}{84LB_m\beta^{L-1}\sqrt{b\ln(4bL)}},$$

*and $\beta$ is any constant satisfying $|\tilde{\beta} - \beta| \leq \frac{\tilde{\beta}}{L}$, then with respect to the random draw of $\{U_l\}_{l=1}^L$,*

$$\Pr\left(\max_{i \in V_0 \cup V_m} \|f(g_i(X,G); \{\widetilde{W}_l\}_{l=1}^L) - f(g_i(X,G); \{\widetilde{W}_l + U_l\}_{l=1}^L)\|_\infty < \frac{\gamma}{8}\right) > \frac{1}{2}.$$

Then we prove Theorem 3 (re-stated as Theorem 6 below).

**Theorem 6** (Subgroup Generalization Bound for GNNs). *Let $\tilde{h}$ be any classifier in $\mathcal{H}$ with parameters $\{\widetilde{W}_l\}_{l=1}^L$. Under Assumptions 1, 2, 3, and 4, for any $0 < m \leq M$, $\gamma \geq 0$, and large enough $N_0$, with probability at least $1 - \delta$ over the sample of $y^0$, we have*

$$\mathcal{L}_m^0(\tilde{h}) \leq \widehat{\mathcal{L}}_0^\gamma(\tilde{h}) + O\left(cK\epsilon_m + \frac{b\sum_{l=1}^L \|\widetilde{W}_l\|_F^2}{(\gamma/8)^{2/L} N_0^\alpha}(\epsilon_m)^{2/L} + \frac{1}{N_0^{1-2\alpha}} + \frac{1}{N_0^{2\alpha}} \ln \frac{LC(2B_m)^{1/L}}{\gamma^{1/L}\delta}\right). \tag{16}$$

*Proof.* There are two main steps in the proof. In the first step, for a given constant $\beta > 0$, we first define the "prior" $P$ and the "posterior" $Q$ on $\mathcal{H}$ in a way complying the conditions in Lemma 1 and Lemma 7. Then for all classifiers with parameters satisfying $|\tilde{\beta} - \beta| \leq \frac{\tilde{\beta}}{L}$, where $\tilde{\beta} = \left(\prod_{l=1}^L \|\widetilde{W}_l\|_2\right)^{1/L}$, we can derive a generalization bound by applying Theorem 2 and Lemma 1. In the second step, we investigate the number of $\beta$ we need to cover all possible relevant classifier parameters and apply a union bound to get the final bound. The second step is essentially the same as Neyshabur et al. [30] while the first step differs by the need of incorporating Lemma 1.

We first show the first step. Given a choice of $\beta$ independent of the training data, let

$$\sigma = \min\left(\frac{(\gamma/8\epsilon_m)^{1/L}}{\sqrt{2b(\lambda N_0^{-\alpha} + \ln 2bL)}}, \frac{\gamma}{84LB_m\beta^{L-1}\sqrt{b\ln(4bL)}}\right).$$

Assume the "prior" $P$ on $\mathcal{H}$ is defined by sampling the vectorized MLP parameters from $\mathcal{N}(0, \sigma^2 I)$; and the "posterior" $Q$ on $\mathcal{H}$ is defined by first sampling a set of random perturbations $\{U_l\}_{l=1}^L$ with $\mathrm{vec}(\{U_l\}_{l=1}^L) \sim \mathcal{N}(0, \sigma^2 I)$ and then adding them to $\{\widetilde{W}_l\}_{l=1}^L$, the parameters of $\tilde{h}$. Then for any $\tilde{h}$ with $\{\widetilde{W}_l\}_{l=1}^L$ satisfying $|\tilde{\beta} - \beta| \leq \frac{\tilde{\beta}}{L}$, by Lemma 7, we have

$$\Pr_{h \sim Q}\left(\max_{i \in V_0 \cup V_m} |h_i(X,G) - \tilde{h}_i(X,G)|_\infty < \frac{\gamma}{8}\right) > \frac{1}{2}.$$

Therefore, by applying Theorem 2, we know the bound (4) holds for $\tilde{h}$, i.e., with probability at least $1 - \delta$,

$$\mathcal{L}_m^0(\tilde{h}) - \widehat{\mathcal{L}}_0^\gamma(\tilde{h})$$
$$\leq \frac{1}{\lambda}\left(2(D_{\mathrm{KL}}(Q\|P) + 1) + \ln\frac{1}{\delta} + \frac{\lambda^2}{4N_0} + D_{m,0}^{\gamma/2}(P;\lambda)\right)$$
$$\leq \frac{1}{\lambda}\left(2(D_{\mathrm{KL}}(Q\|P) + 1) + \ln\frac{1}{\delta} + \frac{\lambda^2}{4N_0} + \ln 3 + \lambda cK\epsilon_m\right) \tag{17}$$
$$\leq \frac{2}{N_0^{2\alpha}}D_{\mathrm{KL}}(Q\|P) + \frac{1}{N_0^{2\alpha}}\left(\ln\frac{3}{\delta} + 2\right) + \frac{1}{4N_0^{1-2\alpha}} + cK\epsilon_m, \tag{18}$$

where in (17) we have applied Lemma 1 to bound $D_{m,0}^{\gamma/2}(P;\lambda)$ under Assumptions 1, 2, and 3; and in (18) we have set $\lambda = N_0^{2\alpha}$.

Moreover, since both $P$ and $Q$ are normal distributions, we know that

$$D_{\mathrm{KL}}(Q\|P) \leq \frac{\sum_{l=1}^L \|\widetilde{W}_l\|_F^2}{2\sigma^2}.$$

By Assumption 4, both $B_m$ and $C$ are constant with respect to $N_0$. Later we will show that we only need $\beta \leq C$. Therefore, for large enough $N_0$, we can have

$$\frac{(\gamma/8\epsilon_m)^{1/L}}{\sqrt{2b(N_0^\alpha + \ln 2bL)}} < \frac{\gamma}{84LB_m\beta^{L-1}\sqrt{b\ln(4bL)}},$$

which implies,

$$\sigma = \frac{(\gamma/8\epsilon_m)^{1/L}}{\sqrt{2b(N_0^\alpha + \ln 2bL)}},$$

and hence,

$$D_{\mathrm{KL}}(Q\|P) \le \frac{b(N_0^\alpha + \ln 2bL)\sum_{l=1}^L \|\widetilde{W}_l\|_F^2}{(\gamma/8)^{2/L}}(\epsilon_m)^{2/L}. \tag{19}$$

Therefore, with probability at least $1-\delta$,

$$
\begin{aligned}
&\mathcal{L}_m^0(\tilde{h}) - \widehat{\mathcal{L}}_0^\gamma(\tilde{h}) \\
&\le cK\epsilon_m + \frac{2}{N_0^{2\alpha}}D_{\mathrm{KL}}(Q\|P) + \frac{1}{N_0^{2\alpha}}\left(\ln\frac{3}{\delta}+2\right) + \frac{1}{4N_0^{1-2\alpha}} \\
&\le O\left(cK\epsilon_m + \frac{b\sum_{l=1}^L\|\widetilde{W}_l\|_F^2}{(\gamma/8)^{2/L}N_0^\alpha}(\epsilon_m)^{2/L} + \frac{1}{N_0^{1-2\alpha}} + \frac{1}{N_0^{2\alpha}}\ln\frac{1}{\delta}\right). 
\end{aligned}
\tag{20}
$$

Then we show the second step, i.e., finding out the number of $\beta$ we need to cover all possible relevant classifier parameters. Similarly as Neyshabur et al. [30], we will show that we only need to consider $(\frac{\gamma}{2B_m})^{1/L} \le \tilde{\beta} \le C$ (recall that $\|\widetilde{W}_l\|_F \le C, l = 1, \dots, L$). For any $\tilde{\beta}$ outside this range, the bound (16) automatically holds. If $\tilde{\beta} < (\frac{\gamma}{2B_m})^{1/L}$, then for any node $i \in V_0$, $\|\tilde{h}_i(X, G)\|_\infty < \frac{\gamma}{2}$, which implies $\widehat{\mathcal{L}}_0^\gamma(\tilde{h}) = 1$ as the difference between any two output logits for any training node is smaller than $\gamma$. Also noticing that $\mathcal{L}_m^0(\tilde{h}) \le 1$ by definition, so the bound (16) trivially holds. And for $\tilde{\beta}$ in this range, $|\beta - \tilde{\beta}| \le \frac{1}{L}(\frac{\gamma}{2B_m})^{1/L}$ is a sufficient condition for $\beta$ to satisfy $|\tilde{\beta} - \beta| \le \frac{\tilde{\beta}}{L}$, and we need at most $\frac{LC(2B_m)^{1/L}}{\gamma^{1/L}}$ of $\beta$ to cover all $\tilde{\beta}$ in the above range. Taking a union bound on all such $\beta$, which is equivalent to replace $\delta$ with $\frac{\delta}{\frac{LC(2B_m)^{1/L}}{\gamma^{1/L}}}$ in (20), it gives us the final result: with probability at least $1-\delta$,

$$\mathcal{L}_m^0(\tilde{h}) - \widehat{\mathcal{L}}_0^\gamma(\tilde{h}) \le O\left(cK\epsilon_m + \frac{b\sum_{l=1}^L\|\widetilde{W}_l\|_F^2}{(\gamma/8)^{2/L}N_0^\alpha}(\epsilon_m)^{2/L} + \frac{1}{N_0^{1-2\alpha}} + \frac{1}{N_0^{2\alpha}}\ln\frac{LC(2B_m)^{1/L}}{\gamma^{1/L}\delta}\right).$$

$\square$

## A.5   Discussion on Assumption 3

To better understand Assumption 3, we show a simplified scenario where this assumption holds.

We discuss in the context where the classification problem has binary labels and the MLP of the classifier $h$ only consists of a linear layer with parameters $W \in \mathbb{R}^{D'\times 2}$. In this case, the distribution $P$ on $\mathcal{H}$ in Assumption 3 is defined by sampling the vectorized parameters $\mathrm{vec}(W) \sim \mathcal{N}(0, \sigma^2 I_{2D'})$. Under Assumption 2, each training sample in $V_0$ has a near set in $V_m$ with the same size $s_m$. For simplicity, we consider the case where $s_m = 1$. Let $Z^{(0)}, Z^{(m)} \in \mathbb{R}^{N_0 \times D'}$ be the aggregated node features of $V_0$ and $V_m$ respectively. Without loss of generality, assume for each $i = 1, \dots, N_0$, the closest point in $Z^{(0)}$ for $Z_i^{(m)}$ is $Z_i^{(0)}$. To simplify the notations, we define $Z := Z^{(0)}$ and $\varepsilon := Z^{(m)} - Z^{(0)}$. We always treat $M_i$ for any matrix $M$ as the transpose of the $i$-th row of $M$ and define $M_{(i)}$ as the $i$-th column vector of $M$.

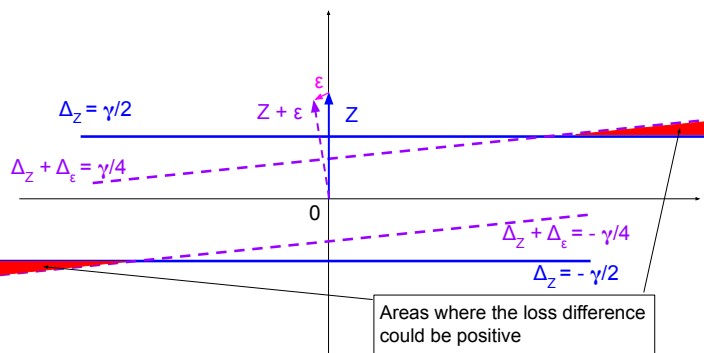

Figure 6: An illustrative example of areas in the space of $\Delta_W$ where the loss difference term for an index $i$ could be positive. For visual simplicity in the figure, we have used $Z$ and $\varepsilon$ to represent $Z_i$ and $\varepsilon_i$.

Following the proof of Lemma 5, and in particular, Eq. (13), it is easy to show that, for any $h \in \mathcal{H}$ with parameters $W$,

$$\mathcal{L}_m^{\gamma/4}(h) - \mathcal{L}_0^{\gamma/2}(h)$$

$$\leq \frac{1}{N_0} \sum_{i=1}^{N_0} \sum_{k=1}^{2} \eta_k(Z_i^{(m)}) \left( \mathcal{L}^{\gamma/4}((Z^{(m)} \cdot W)_i, k) - \mathcal{L}^{\gamma/2}((Z^{(0)} \cdot W)_i, k) \right) + c\epsilon_m$$

$$= 2c\epsilon_m + \frac{1}{N_0} \sum_{i=1}^{N_0} \sum_{k=1}^{2} \eta_k(Z_i^{(m)}) \left( \mathbb{1}\left[ W_{(k)}^T(Z_i + \varepsilon_i) < \frac{\gamma}{4} + W_{(3-k)}^T(Z_i + \varepsilon_i) \right] - \mathbb{1}\left[ W_{(k)}^T Z_i < \frac{\gamma}{2} + W_{(3-k)}^T Z_i \right] \right).$$
(21)

For Assumption 3 to hold, a sufficient condition is to have the second term in Eq. (21) smaller than $N^\alpha$ for any $h$. Below we will investigate when this sufficient condition holds.

To further simplify the notations, we define $\Delta_W := W_{(1)} - W_{(2)}$, $\Delta_Z := Z\Delta_W$, $\Delta_\varepsilon := \varepsilon\Delta_W$, and $\eta_k^i := \eta_k(Z_i^{(m)})$. Then

$$\mathcal{L}_m^{\gamma/4}(h) - \mathcal{L}_0^{\gamma/2}(h)$$

$$\leq 2c\epsilon_m + \frac{1}{N_0} \sum_{i=1}^{N_0} \sum_{k=1}^{2} \eta_k^i \left( \mathbb{1}\left[ (-1)^{k+1}(\Delta_Z + \Delta_\varepsilon)_i < \frac{\gamma}{4} \right] - \mathbb{1}\left[ (-1)^{k+1}\Delta_{Zi} < \frac{\gamma}{2} \right] \right). \quad (22)$$

Note that since $\text{vec}(W) \sim \mathcal{N}(0, \sigma^2 I_{2D'})$, we have $\Delta_W \sim \mathcal{N}(0, 2\sigma^2 I_{D'})$. And the second term in (22) depends on $W$ only through $\Delta_W$.

Table 1: Possible values of the loss difference term for each index $i = 1, \ldots, N_0$.

| | $\Delta_{Zi} > \frac{\gamma}{2}$ | $\Delta_{Zi} < -\frac{\gamma}{2}$ | $-\frac{\gamma}{2} \leq \Delta_{Zi} \leq \frac{\gamma}{2}$ |
|---|---|---|---|
| $(\Delta_Z + \Delta_\varepsilon)_i > \frac{\gamma}{4}$ | $0$ | $\eta_2^i - \eta_1^i$ | $-\eta_1^i$ |
| $(\Delta_Z + \Delta_\varepsilon)_i < -\frac{\gamma}{4}$ | $\eta_1^i - \eta_2^i$ | $0$ | $-\eta_2^i$ |
| $-\frac{\gamma}{4} \leq (\Delta_Z + \Delta_\varepsilon)_i \leq \frac{\gamma}{4}$ | $\eta_1^i$ | $\eta_2^i$ | $0$ |

For each $i = 1, \ldots, N_0$, the term $\sum_{k=1}^{2} \eta_k^i \left( \mathbb{1}\left[ (-1)^{k+1}(\Delta_Z + \Delta_\varepsilon)_i < \frac{\gamma}{4} \right] - \mathbb{1}\left[ (-1)^{k+1}\Delta_{Zi} < \frac{\gamma}{2} \right] \right)$ in (22) has only a few possible values, which can be summarized in the following 9 cases in Table 1. As can be seen, this loss difference term could be positive only when (1) $\Delta_{Zi} > \frac{\gamma}{2}$ and $(\Delta_Z + \Delta_\varepsilon)_i \leq \frac{\gamma}{4}$ or (2) $\Delta_{Zi} < -\frac{\gamma}{2}$ and $(\Delta_Z + \Delta_\varepsilon)_i \geq -\frac{\gamma}{4}$. This implies that, for fixed $Z_i$ and $\varepsilon_i$, there are two linear subspaces in the space of $\Delta_W$ where the loss difference for index $i$ could be positive. In Figure 6, we provide an illustrative example of such linear subspaces in the case $\Delta_W \in \mathbb{R}^2$, such that we can visualize it. Qualitatively, when $\|\varepsilon_i\|_2$ is much smaller than $\|Z_i\|_2$ (which is often the case by their constructions), the areas that the loss difference term being positive will be very small.

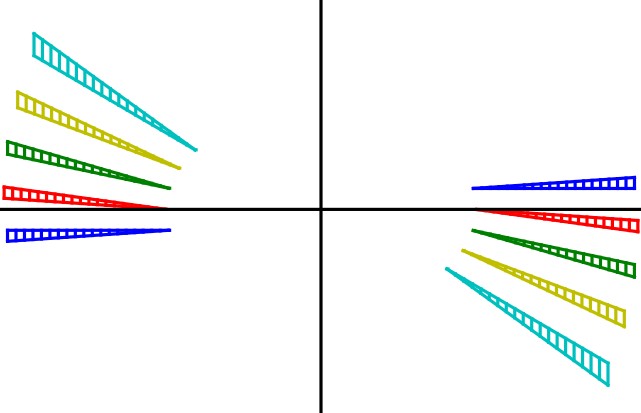

Figure 7: An illustrative example of data points with no intersected positive areas. Only the positive areas as shown in Figure 6 are visualized. Each color corresponds to a unique index $i$. As shown in the figure, there are no intersections among the areas of different data points when the $Z_i$'s are nicely scattered and $\varepsilon_i$'s are small.

For a classifier $h$ to have $\mathcal{L}_m^{\gamma/4}(h) - \mathcal{L}_0^{\gamma/2}(h) > cK\epsilon_m + N_0^{-\alpha}$, a necessary condition is that its corresponding $\Delta_W$ lies in the intersection of positive areas of at least $N_0^{1-\alpha}$ samples. Conversely, if $\varepsilon_i$'s are small and $Z_i$'s are nicely scattered such that the $N_0$ samples can be divided into $N_0^\alpha$ groups where the positive areas of any two points from different groups do not intersect, then we know $\mathcal{L}_m^{\gamma/4}(h) - \mathcal{L}_0^{\gamma/2}(h) \le cK\epsilon_m + N_0^{-\alpha}$ for any $h$. And hence this is a sufficient condition for Assumption 3 to hold. Figure 7 provides an illustrative example of data points with no intersected positive areas on a 2-dimensional surface. When $D' > 2$, it might be difficult to completely avoid intersections of the positive areas. However, what Assumption 3 requires is that the areas where a large number of data points intersect are small.

# B More Details of Experiment Setup

In this section, we describe more details of our experiment setup that are omitted in the main paper due to space limit.

## B.1 Detailed Training Setup

We use the default setting in Deep Graph Library [43][12] for model hyper-parameters. We use the Adam optimizer with initial learning rate of 0.01 and weight decay of 5e-4 to train all models for 400 epoch by minimizing the cross entropy loss, with early stopping on the validation set.

## B.2 Detailed Setup of the Noisy Feature Experiment

In this experiment (corresponding to Figure 4), we make the node features less homophilious by adding random noises to each node independently. Specifically, we use noisy features $\tilde{X} = X + \alpha \frac{\|X\|_F}{\|U\|_F} U$, where $X \in \mathbb{R}^{N \times D}$ is the original feature matrix, and $U \in \mathbb{R}^{N \times D}$ is a random matrix with each element independently and uniformly sampled from $[0, 1]$. And we set $\alpha = 5$. In this way, the magnitude of the noise is slightly larger than the original feature to significantly reduce the homophily property. All other experiment settings are the same as those corresponding to Figure 1.

---

[12] Apache License 2.0.

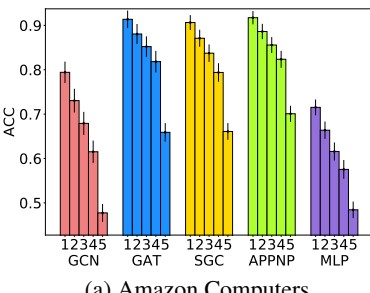 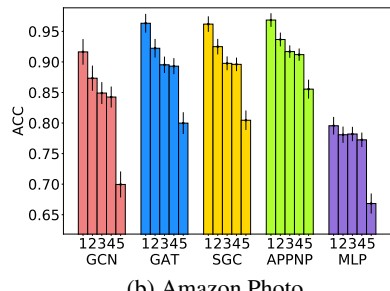

(a) Amazon Computers.        (b) Amazon Photo.

Figure 8: Test accuracy disparity across subgroups by aggregated-feature distance. Extra experiments on Amazon-Computers and Amazon-Photo datasets. The experiment and plot settings are the same as Figure 1.

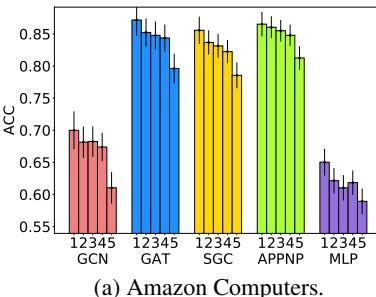 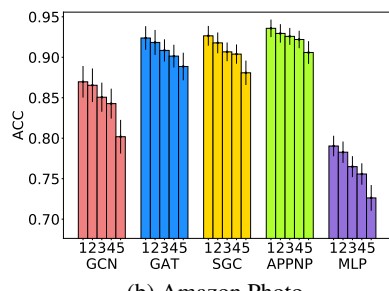

(a) Amazon Computers.        (b) Amazon Photo.

Figure 9: Test accuracy disparity across subgroups by geodesic distance. Extra experiments on Amazon-Computers and Amazon-Photo datasets. The experiment and plot settings are the same as Figure 2.

### B.3   Detailed Setup of the Biased Training Node Selection Experiment

In this experiment (corresponding to Section 5.2), we investigate the impact of biased training node selection. As briefly described in Section 5.2, we choose a "dominant class" and construct a manipulated training set. For each class, we still sample 20 training nodes but in a biased way. Specifically, given one choice of the four node centrality metrics (degree, closeness, betweeness, and PageRank), the training set is sampled as follows.

1. For the dominant class, uniformly sample 15 nodes from the 10% of the nodes with highest node centrality, and uniformly sample 5 nodes from the remaining.

2. For each of the other classes, uniformly sample 15 nodes from the 10% of the nodes with lowest node centrality, and uniformly sample 5 nodes from the remaining.

In this way, the training nodes of the dominant class are biased towards high-centrality nodes while the training nodes of the other classes are biased towards low-centrality nodes.

After the biased training set is constructed, we randomly sample 500 validation nodes and 1000 test nodes from the remaining nodes and perform the model training following the standard setup as the previous experiments.

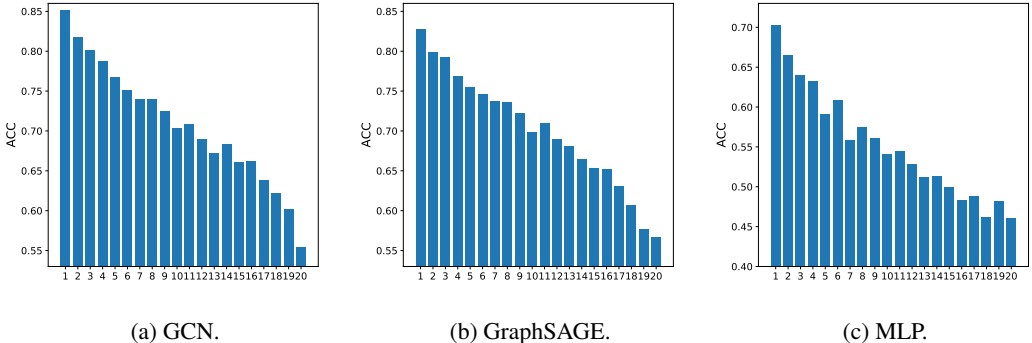

| (a) GCN. | (b) GraphSAGE. | (c) MLP. |

Figure 10: Results on OGBN-Arxiv. Test accuracy disparity across subgroups by aggregated-feature distance. Each figure corresponds to a model. Bars labeled 1 to 20 represent subgroups with increasing distance to training set.

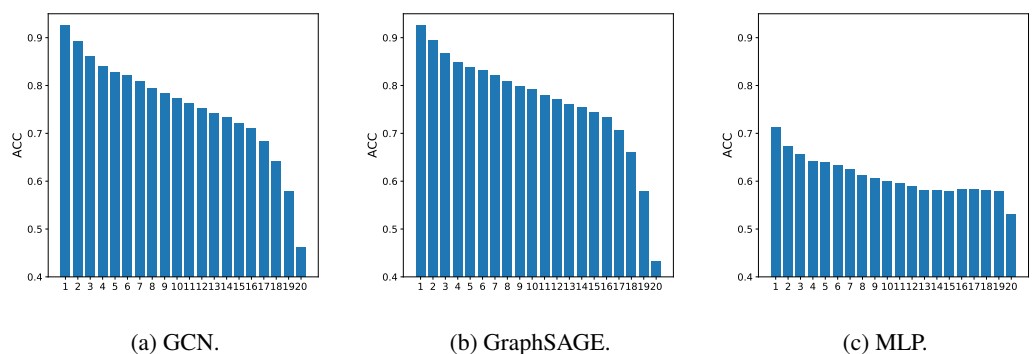

| (a) GCN. | (b) GraphSAGE. | (c) MLP. |

Figure 11: Results on OGBN-Products. Test accuracy disparity across subgroups by aggregated-feature distance. The experiment and plot settings are the same as Figure 10.

## C  Extra Experiment Results

### C.1  Accuracy Disparity on Amazon Datasets

In addition to the commonly used citation network benchmarks, Cora, Citeseer, and Pubmed [37, 47], we also provide results of the test accuracy disparity experiments of subgroups by aggregated-feature distance and geodesic distance on Amazon-Computers and Amazon-Photo datasets [38], which have a similar scale but a different network type compared to the three citation networks.

For Amazon-Computers and Amazon-Photo, we follow exact the same experiment procedure as for Cora, Citeseer, and Pubmed. The results of subgroups by aggregated-feature distance are shown in Figure 8 and the results of subgroups by geodesic distance are shown in Figure 9. The results are respectively similar as those in Figure 1 and Figure 2.

### C.2  Accuracy Disparity on Open Graph Benchmarks

We further provide experiment results on two large-scale datasets from Open Graph Benchmark [17], OGBN-Arxiv and OGBN-Products.

For OGBN-Arxiv and OGBN-Products, we first follow the standard training procedure suggested by Open Graph Benchmark [17] to train a GCN, a GraphSAGE, and an MLP. And we split the test groups into 20 groups in terms of the aggregated feature distance. As there are more test nodes available, we can afford the split of more groups better resolution. The results on OGBN-Arxiv and OGBN-

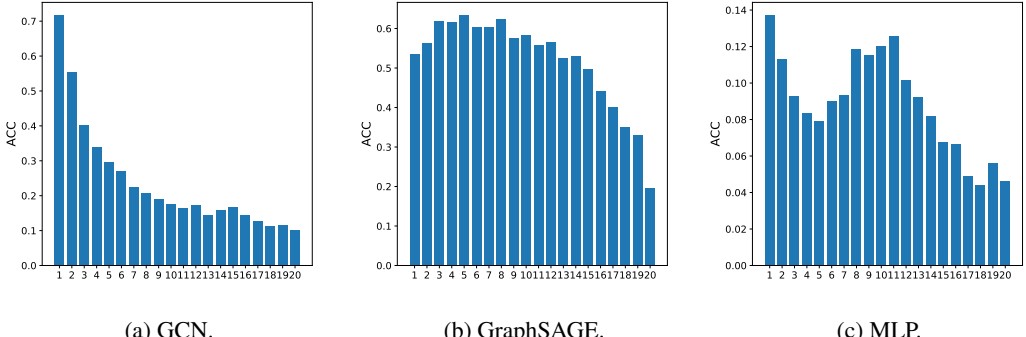

Figure 12: Results on OGBN-Arxiv. Test accuracy disparity across subgroups by aggregated-feature distance, experimented with noisy features. The experiment and plot settings are the same as Figure 10, except for the node features are perturbed by independent noises to reduce homophily.

Products are respectively shown in Figure 10 and Figure 11, where we observe a similar decreasing pattern of test accuracy as in Figure 1 (on the citation networks). Since there is also a decreasing pattern for MLP, following the experiments shown in Figure 4, we further inject independent noises to node features to reduce the homophily of the OGBN-Arxiv dataset and repeat the experiments in Figure 10. The results are shown in Figure 12, where, similar as Figure 4, the decreasing pattern largely remains for GNNs but disapears for the MLP.

We also experiment on subgroups split in terms of geodesic distance and node centrality metrics. The results of these experiments are slightly different on the large-scale datasets compared to those on the smaller benchmark datasets.

For geodesic distance (Figure 13 for OGBN-Arxiv and Figure 14 for OGBN-Products), there is not a descending trend of test accuracy until the last few groups. This is because the size of training set is large such that most test nodes are 1-hop neighbors of some training nodes. Therefore most groups are random split of such 1-hop neighbors and there will not be a descending accuracy among these subgroups. This problem is especially obvious for OGBN-Arxiv, where 60% of the nodes are in the training set. So we only see accuracy drop on the last two subgroups. The size of training set is relatively smaller on OGBN-Products but still more than 60% of the test nodes are 1-hop neighbors of some training nodes. It is worth-noting that the plots in Figure 14 have stair patterns, showing a clear descending trend with respect to the geodesic distance. The fluctuation of early subgroups is larger on OGBN-Arxiv because there are fewer test nodes in OGBN-Arxiv than in OGBN-Products. Overall, there is still a clear descending trend with respect to increasing geodesic distance. But the nodes are less distinguishable in terms of geodesic distance than aggregated-feature distance, especially when the size of training set is large (more discussions in Appendix D.1).

For node centrality metrics, we report experiments on degree and PageRank, and omit the betweenness and closeness metrics due to their high computation cost on large-scale graphs. The results on OGBN-Arxiv are shown in Figure 15. It is intriguing that there is a descending trend with respect to the degree and PageRank metrics on this particular experiment setting, though the descending trend is not as sharp as the one in Figure 10 (experiments on aggregated-feature distance). It is possible that, when there is a very large training set (60% in this case), the node centrality metrics become related to the aggregated-feature distance. However, node centrality metrics again fail to capture the descending trend on OGBN-Products, as shown in Figure 16. In future work, we plan to further explore the relationship between the theoretically derived aggregated-feature distance and various more intuitive graph metrics on different graph data.

### C.3 More Results of Biased Training Node Selection

In Figure 5 of Section 5.2, we have shown that the learned GNN models will be biased towards the labels of training nodes of higher centrality (while the learned MLP models do not show a similar trend). Due to space limit, we are only able to report the experiment results on Cora with a

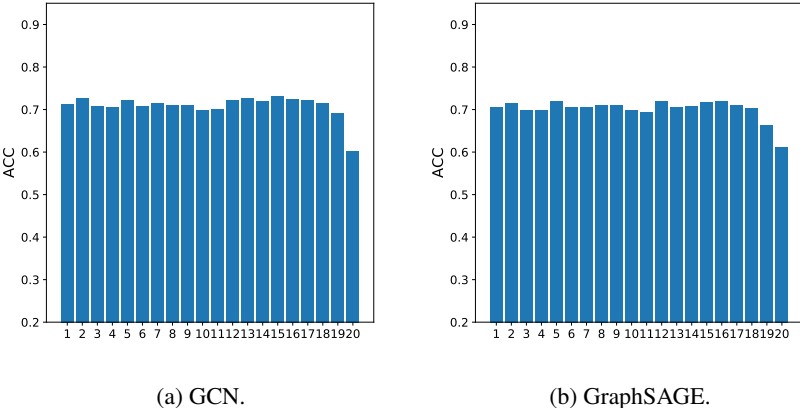

(a) GCN.

(b) GraphSAGE.

Figure 13: Results on OGBN-Arxiv. Test accuracy disparity across subgroups by geodesic distance. The experiment and plot settings are the same as Figure 10, except for the aggregated-feature distance is replaced by the geodesic distance.

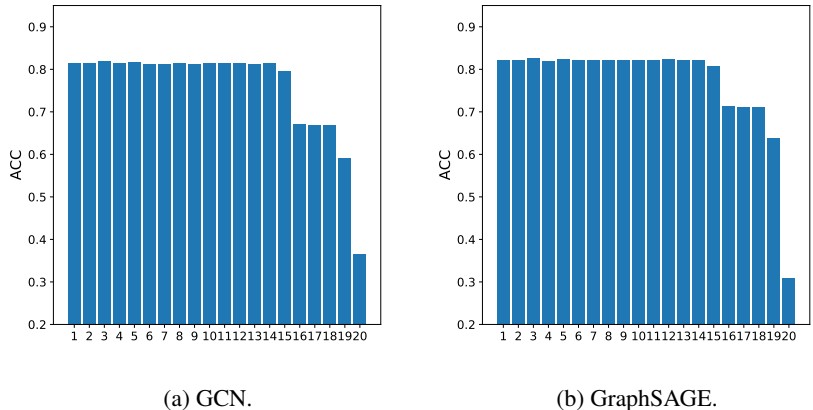

(a) GCN.

(b) GraphSAGE.

Figure 14: Results on OGBN-Products. Test accuracy disparity across subgroups by geodesic distance. The experiment and plot settings are the same as Figure 13.

particular class selected as the "dominant" class. Here we report the full experiment results on three datasets, with each class selected as the "dominant" class. The results on Cora, Citeseer, and Pubmed are respectively shown in Figures 17, 18, and 19. As can be seen from the figures, the observed phenomenon is consistent over almost all settings.

## D  Discussions

### D.1  Relationship Between Aggregated-Feature Distance and Geodesic Distance

We discuss two scenarios where the aggregated-feature distance and the geodesic distance are likely to be related.

*Smoothing effect of feature aggregation in GNNs.* Many existing GNN models are known to have a smoothing effect on the aggregated node features [24]. As a result, nodes with a shorter geodesic distance are likely to have more similar aggregated features.

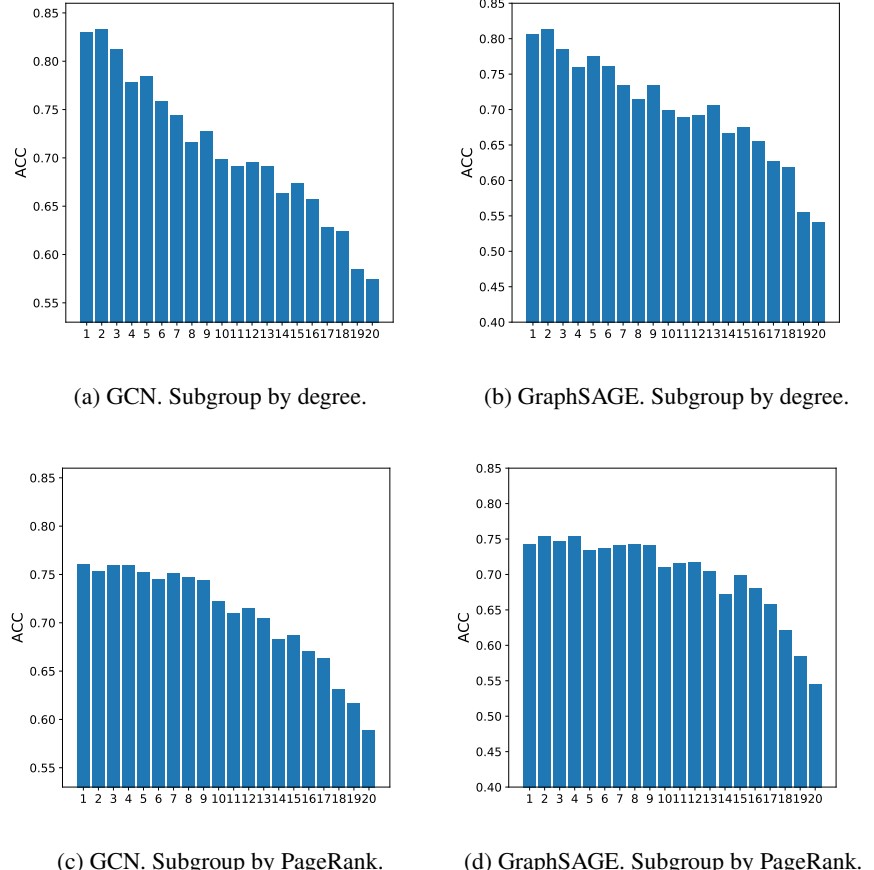

(a) GCN. Subgroup by degree.

(b) GraphSAGE. Subgroup by degree.

(c) GCN. Subgroup by PageRank.

(d) GraphSAGE. Subgroup by PageRank.

Figure 15: Results on OGBN-Arxiv. Test accuracy disparity across subgroups by node centrality metrics. The experiment and plot settings are the same as Figure 10.

*Homophily.* Many real-world graph-structured data exhibit a homophily property [28], i.e., connected nodes tend to share similar attributes. In this case, again, nodes with a shorter geodesic distance on the graph tend to have more similar aggregated features.

However, the geodesic distance is usually coarser-grained than the aggregated-feature distance due to its discrete nature. When the graph is a "small world" [44] and the number of training nodes is large, the geodesic distance from most test nodes to the set of training nodes will concentrate on 1 or 2 hops, making the test nodes indistinguishable with respect to this metric.

It is an interesting future direction to explore interpretable graph metrics that may better relate to the aggregated-feature distance.

## D.2 Implications for GNN Generalization under Non-Homophily

A number of recent studies suggest that classical GNNs (e.g., GCN [20]) can only work well when the labels of connected nodes are similar [31, 16, 52], which is now commonly referred as homophily [28]. However, homophily is not a necessary condition to have small generalization errors in our analysis. Instead, good generalization can be achieved when the aggregated features of test nodes are close to those of some training nodes. Interestingly, a concurrent work [26] of this paper observes a similar phenomenon with empirical evidence.

The new results by Ma et al. [26] and our work suggest that the space of non-homophilious data can be further dissected into more fine-grained categories, which may motivate designs of new GNN models tailored for each category.

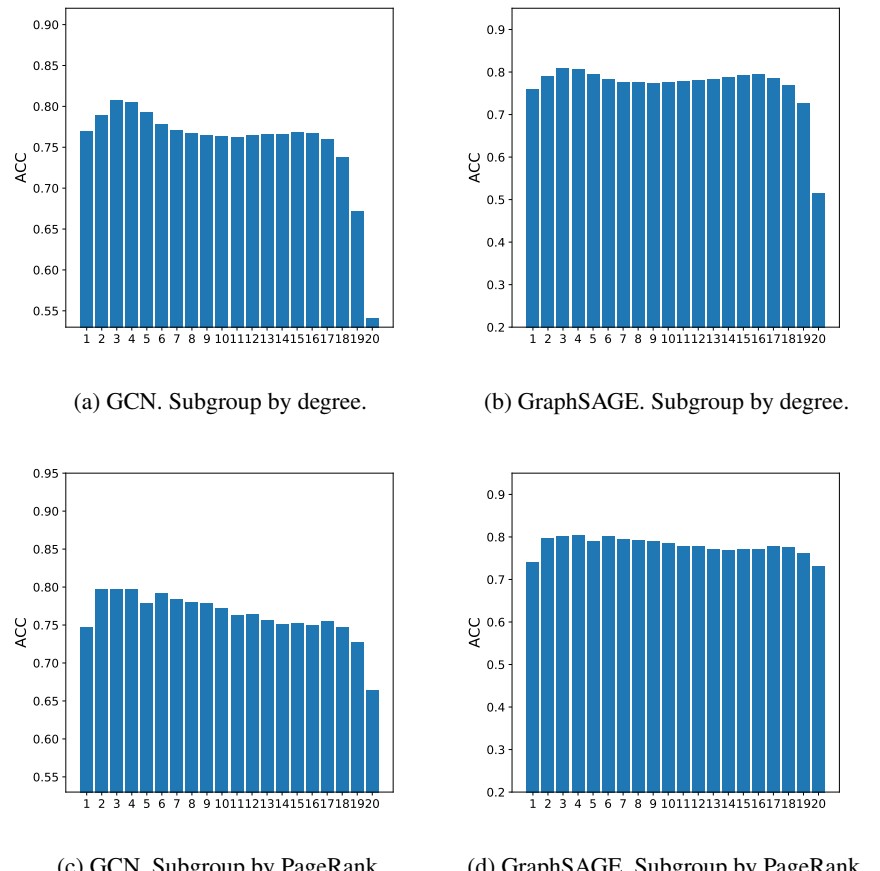

(a) GCN. Subgroup by degree.

(b) GraphSAGE. Subgroup by degree.

(c) GCN. Subgroup by PageRank.

(d) GraphSAGE. Subgroup by PageRank.

Figure 16: Results on OGBN-Products. Test accuracy disparity across subgroups by node centrality metrics. The experiment and plot settings are the same as Figure 11.

### D.3 Limitations of the Analysis

To our best knowledge, this work is one of the first attempts[13] to theoretically analyze the generalization ability of GNNs under non-IID node-level semi-supervised learning tasks. While we believe this work presents non-trivial contributions towards the theoretical understanding of generalization and fairness of GNNs with supportive empirical evidences, there are a few limitations of the current analysis which we hope to improve in future work.

The first limitation is that the derived generalization bounds do not yet match the practical performances of GNNs. This limitation is partly inherited from the mismatch between the theories and the practices of deep learning in general, as we utilize the results by Neyshabur et al. [30] to illustrate the characteristics of the neural-network part of GNNs. In future work, we hope to adapt stronger PAC-Bayesian bounds for neural networks under IID setup [51, 10] to the non-IID setup for GNNs.

Another limitation is that we have assumed a particular form of GNNs similar as SGC [45] or APPNP [21]. This form of GNNs simplifies the analysis but does not include some common GNNs such as GCN [20] and GAT [41]. We notice that the key characteristics of GNNs we need for the analysis is that the change of outputs of GNNs under certain perturbations needs to be bounded. A recent work [25] has shown that some more general forms of GNNs (including GCN) indeed have bounded output changes under perturbations. So the analysis in this work can be potentially adapted to more general forms of GNNs by utilizing such perturbation bounds. Empirically, we have

---

[13]The only other work we are aware of is by Baranwal et al. [3], where strong assumptions (CSBM) on the data generating mechanisms are made.

demonstrated that the accuracy disparity phenomenon predicted by our theoretical analysis indeed appears in experiments on GCN, GAT, and GraphSAGE.

Finally, our analysis requires some assumptions on the relationship between the training set and the target test subgroup. While, not surprisingly, we have to make some assumptions about this relationship to expect good generalization to the target subgroup, it is an interesting future direction to explore more relaxed assumptions than the ones used in this work.

### D.4 Societal Impacts

As GNNs have been deployed in human-related real-world applications such as recommender systems [48], understanding the fairness issues of GNNs may have direct societal impacts. On the positive side, understanding the systematic biases embedded in the GNN models and the graph-structured data helps researchers and practitioners come up with solutions that mitigate the potential harms resulted by such biases. On the negative side, however, such understanding may also be used for malicious purposes: e.g., performing adversarial attacks on GNNs that utilizes systematic biases. Nevertheless, we believe the theoretical understandings resulted from this work contributes to a small step towards making the GNN models more transparent to the research community, which may motivate the design of better and fairer models.

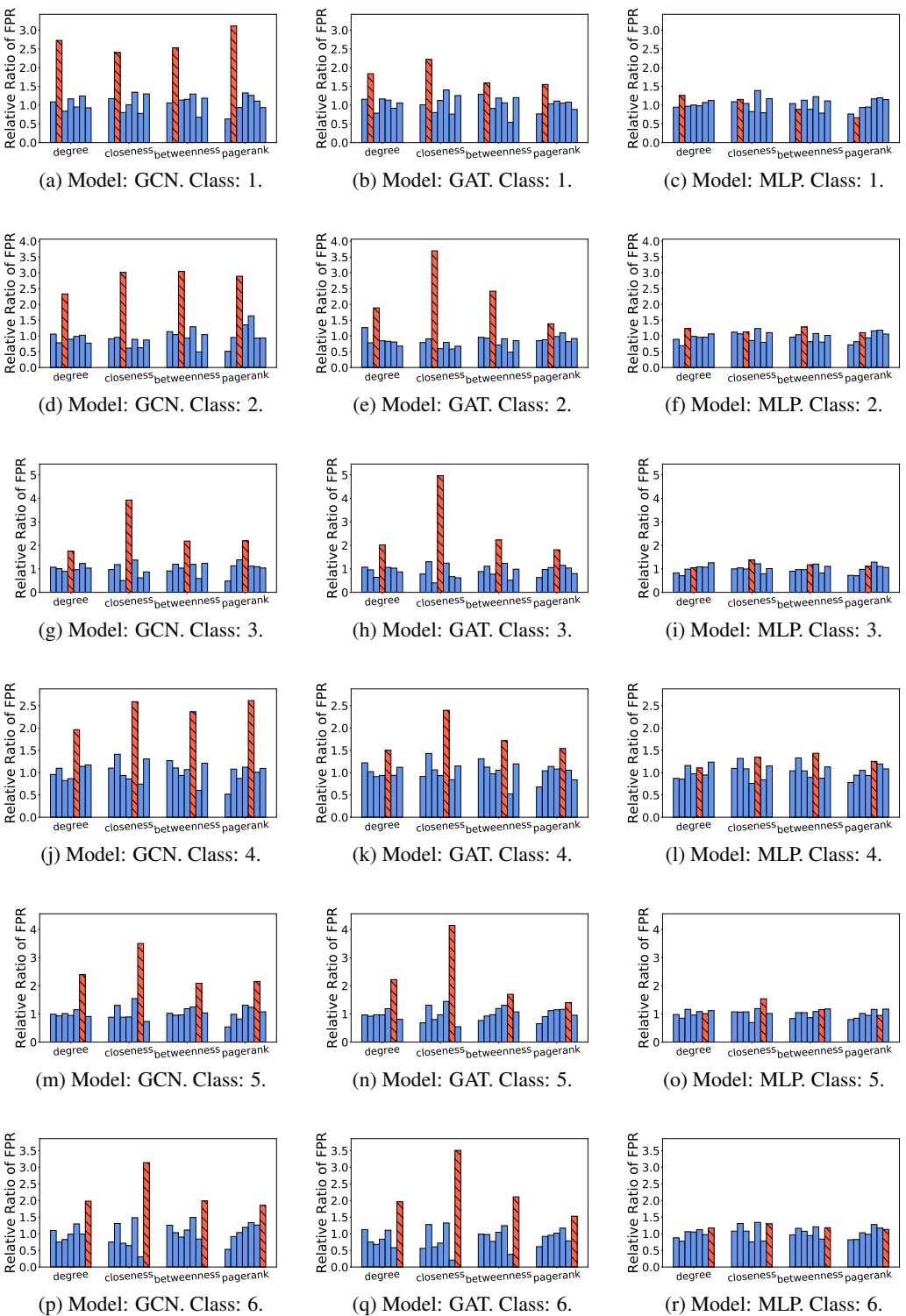

Figure 17: Relative ratio of FPR in the biased training node selection experiment. Remaining results on Cora besides Figure 5. Each row corresponds to a different dominant class of choice. See Figure 5 for the plot settings.

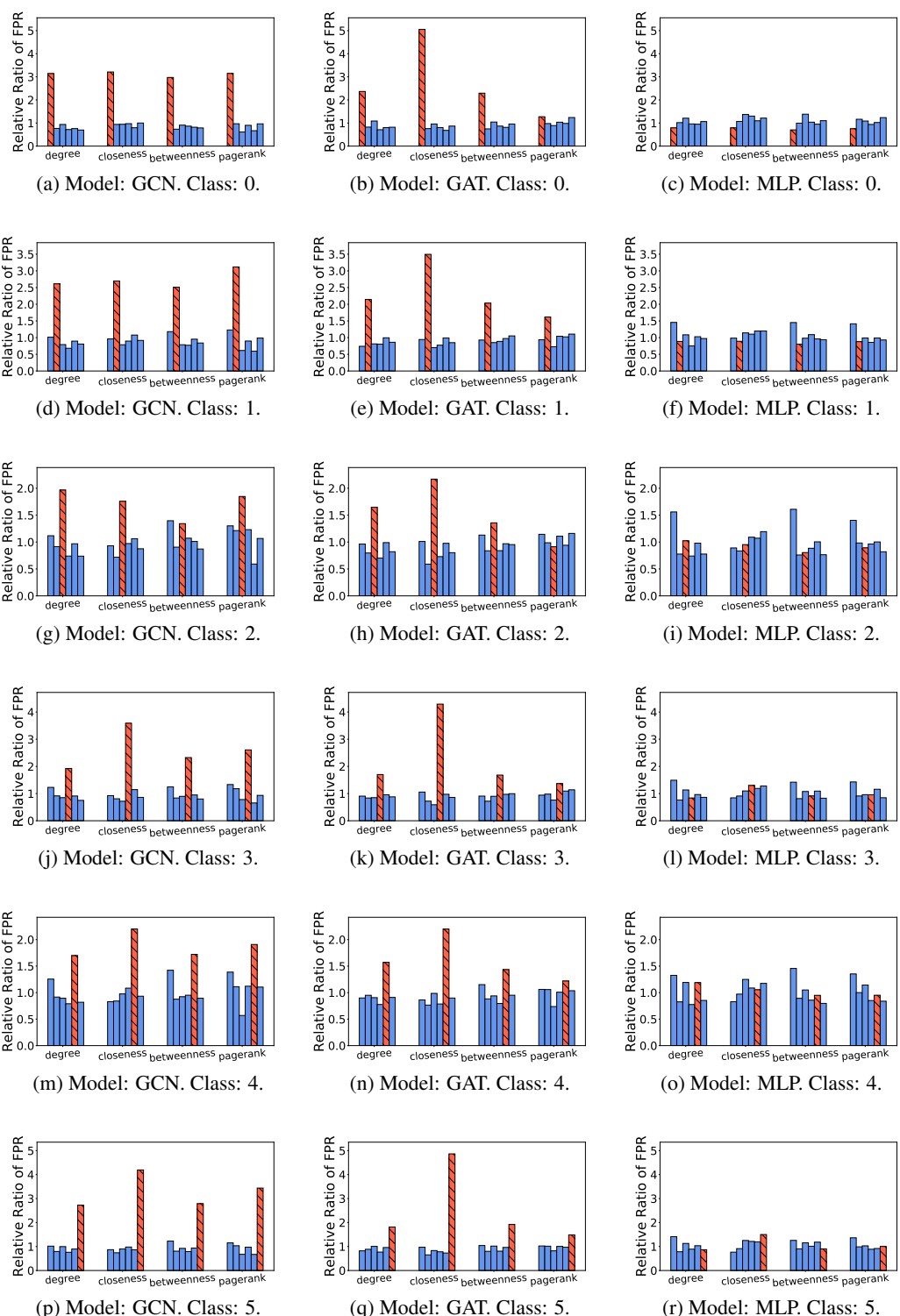

Figure 18: Relative ratio of FPR in the biased training node selection experiment. Full results on Citeseer. Each row corresponds to a different dominant class of choice. See Figure 5 for the plot settings.

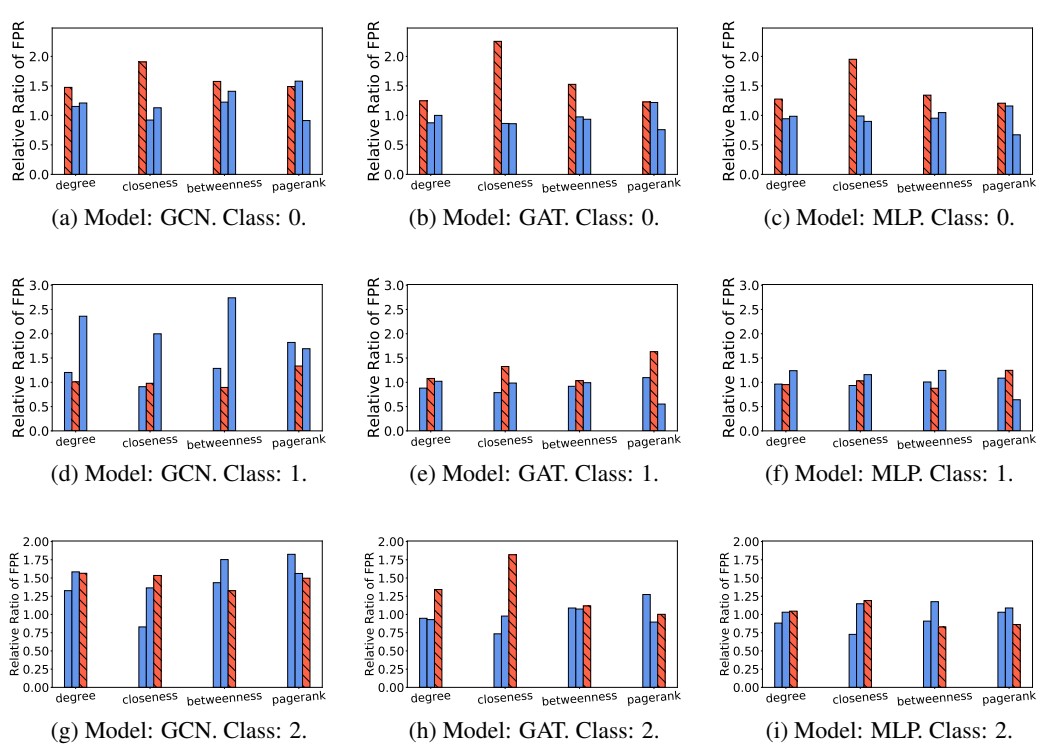

Figure 19: Relative ratio of FPR in the biased training node selection experiment. Full results on Pubmed. Each row corresponds to a different dominant class of choice. See Figure 5 for the plot settings.