# OpenReview forum: "Subgroup Generalization and Fairness of Graph Neural Networks"
_NeurIPS.cc/2021/Conference — NeurIPS 2021 Spotlight_

### Official Review · Reviewer_bVjk · 2021-07-12

**Rating:** 7
**Confidence:** 3

**Summary:**

This work presents PAC-Bayesian analysis for GNNs under a non-IID
semi-supervised learning setup. It first analyzes the generalization performance
for general settings, and then applies the results to GNNs. The theoretical
results have been backed up with empirical evaluations on several graph
benchmarks.


**Limitations And Societal Impact:**

Yes

**Main Review:**

This work presents PAC-Bayesian analysis for GNNs under a non-IID
semi-supervised learning setup. It first analyzes the generalization performance
for general settings, and then applies the results to GNNs. The theoretical
results have been backed up with empirical evaluations on several graph
benchmarks.

The theoretical analysis is solid. It provides new theoretical results for
classification problems in non-IID semi-supervised settings. The implications
and findings from the theoretical and empirical results are also interesting,
which suggest that nodes with larger geodesic distance to the training nodes
suffer from a lower classification accuracy.

However, the main analysis is not specific to graphs and GNNs per se. The
majority of the theoretical work is more about PAC-Bayesian analysis for
subgroups under non-IID semi-supervised setting. Theorem 1 & 2 provide the
generalization bounds for stochastic and deterministic settings, respectively.
Theorem 3 applies Theorem 2 to obtain a subgroup generalization bound for GNNs.
However, the required assumption (Assumption 3) can be strong, for which
real-world graphs may not satisfy. Also, the generalization bound in Equation 6
depends on epsilon_m, how can this term be estimated in practice?

The fairness component of this work is relative weak compared to the
generalization performance analysis. Most of the theoretical contributions are
more devoted to generalization performance under the subgroup assumptions, while
the relationships to fairness is marginal. I suggest not emphasizing the
fairness aspect in the title.


**Time Spent Reviewing:**

4 hours

---

> ### Author Response · Authors · 2021-08-10
> **Response to Reviewer bVjk**
>
> We appreciate the insightful comments by the reviewer.
>
> We acknowledge that some of the assumptions required for the generalization bound for GNNs are not perfectly ideal, which are due to the difficulty of analyzing complicated neural network models. To your great point,  theorems 1 & 2 are indeed not specific to graphs and could be applied to other settings. We apply the general theorems to the analysis of GNNs to strengthen the connection of our theoretical results to practical scenarios, as graph-based learning has been one of the most popular semi-supervised learning setup in recent years, and yet there lacks theoretical understanding on the generalization capability of GNNs. Regarding $epsilon_m$, our empirical study suggests that the aggregated-feature distance could serve as a proxy approximating it.
>
> Regarding the fairness component, we first note that the non-IID setup is necessary for us to theoretically investigate the accuracy disparity (unfairness) problem. Under IID assumption, the generalization capability would have been the same on all test samples by definition. So it is a unique strength of our analysis to analyze the fairness of GNNs, given that non-IID results on GNNs have been rare. Furthermore, our empirical study mostly focuses on the fairness problem, and verifies our hypotheses inspired by the theoretical analysis. Overall, we think there is a tight connection between the generalization analysis and the accuracy disparity phenomenon, with support of extensive empirical evidence.

---

> > ### Comment · Reviewer_bVjk · 2021-08-16
> > **Keep my score**
> >
> > Thanks for the authors' responses. I keep my score after reading the responses. I did not have any major issues to be fixed anyway.

---

> > > ### Author Response · Authors · 2021-08-18
> > > **Thank you!**
> > >
> > > Dear Reviewer bVjk,
> > >
> > > Thanks again for your time and feedback!

---

### Official Review · Reviewer_E1Li · 2021-07-15

**Rating:** 6
**Confidence:** 4

**Summary:**

This paper investigates the subgroup generalization for graph neural networks. In particular, the authors present a PAC-Bayesian analysis for the generalization ability of GNNs on node-level semi-supervised learning tasks. By some theoretical analysis and empirical results, the authors demonstrate the claims to some extent.

**Limitations And Societal Impact:**

Please see the Main Review.

**Main Review:**

This paper considers the subgroup generalization for GNNs, which may potentially influence the performance of different subgroups, thus it may partly contribute to the choice of test set in the experiments of GNNs. However, several main concerns should be addressed by the authors.

1. The paper indeed explores accuracy disparity across subgroups by three subgroup split methods, and through the theoretical analysis and empirical results, the diversity of subgroups may more or less involve some disparity across subgroups. But in my opinion, the investigation of influence on performance by different subgroups is a relatively trivial task, which is an easy thing to think about. In particular, the split of subgroups can be regarded as splitting the nodes by some kind of sensitive attribute. In other words, the proposed distance (such as aggregated-feature distance or geodesic distance) can be viewed as one specific kind of sensitive attribute.
Therefore, though with some theoretical analysis, I still think the main contribution of this paper is limited. That is, this paper first proposes several findings on subgroup generalization for GNNs, and then provides some theoretical analysis to these findings, yet no effective models/suggestions are proposed by the authors to deal with the underlying bias (or unfairness). This makes the paper tend to be a technical report.


2. In lines 109-110 of Sect-3.1, the authors said "Note that the analysis on any ... on the entire unlabeled set, as the entire set is a subset". I'm a bit confused here, and why the reason is "as the entire set is a subset"?

3. In Sect-5.1, the two-step aggregated features are calculated and they can facilitate the computation of distances between nodes. However, I doubt that the two-step aggregated features cannot truly reflect the involved feature information of each node in the two-layers aggregation, since in each GNN layer, the GNN model would filter the aggregated information (such as by transformation, or even attention mechanism to assign different weights) for each node instead of directly aggregating. This may impair the credibility of the results.

After rebuttal: I have read the authors' responses, and suggest the authors make further modifications to the current version to make it more solid.

**Time Spent Reviewing:**

5 hours

---

> ### Author Response · Authors · 2021-08-10
> **Response to Reviewer E1Li**
>
> We thank the reviewer for the questions and comments. And we address your individual concerns below.
>
> 1. We would like to highlight that both the theoretical analysis and the empirical findings of decreasing accuracy patterns are non-trivial.
>
>     (1) For the theoretical analysis, we first note that the generalization analysis for GNNs under the non-IID semi-supervised learning setup is itself novel and non-trivial, and the implications to accuracy disparity are particularly due to the non-IID setup.
>
>     (2) For the empirical findings, it is worth noting the subgroup accuracy with respect to the aggregated-feature distance demonstrates **a decreasing pattern**, while the subgroup accuracy with respect to various node-centrality metrics shows no clear pattern. Furthermore, the findings are **consistent** across multiple GNN models on multiple datasets. This suggests that the accuracy disparity among different subgroups is not random, but instead presents systematic bias towards nodes with certain network properties (i.e., those with small aggregated-feature & geodesic distances to training nodes), and such network properties cannot be captured by commonly used network metrics (e.g., node centrality).
>
> 2. The phrase "the entire set is a subset" simply refers to the fact that, for any set $A$, $A\subseteq A$, so the "entire set" $A$ is a (special) subset of itself. In our analysis, we provided results for any subset of the unlabeled set. The entire unlabeled set is also a subset of itself, so the results also naturally hold for the entire unlabeled set. We apologize for the confusing wording and will revise it in the revised version of the paper.
>
> 3. While we agree that the GNN aggregations are usually more complicated than the way we calculate the aggregated features, our empirical results in fact suggest that we do not need the full GNN information to understand its potential biases. We would argue that this is desired as it implies that the findings are not tied to certain specific GNN models. And it also suggests that there might be general recipes for different kinds of GNNs to mitigate the unfairness problem discovered by this work.

---

> > ### Comment · Reviewer_E1Li · 2021-08-20
> > **Score change to 6**
> >
> > I thank the authors' response, which addressed several of my concerns. I think this is a good paper that may make a good contribution to the research community. I would like to change my score to 6.

---

### Official Review · Reviewer_8vwm · 2021-07-17

**Rating:** 8
**Confidence:** 4

**Summary:**

This paper studies the non-IID sample generalization in Graph Neural Networks in a subgroup PAC-Bayesian analysis framework. Non-IID training samples are widely existing in real-world applications. It first derives several interesting theories based on several reasonable assumption on smoothness and node features. Several key observations are found between different subgroups in aggregated-feature and graph distance space support the theoretical discoveries.

**Limitations And Societal Impact:**

Yes.

**Main Review:**

In my opinion, this is a very good paper on two folds. First, non-IID samples are seen everywhere in the real-world applications, where IID annotations are hard to obtain. Without too many assumptions, the PAC generalization bound brings $\epsilon_m$ into the central role of the disparity or distance between groups. Second, the experiment design specifically constructs different subgroups and reproduce the outcome indicated by the theorem. By comparing Figure 2 and Figure 4, the author use the MLP ablation to reveal the graph homophily in an interesting perspective.

One question:
Is the aggregated feature same as the final hidden representation (before activation) for GCNs? If not, the final hidden representation might be a unified measure for both graph distance and aggregated feature for subgroup.

I believe the proposed analysis framework can guide the fair(unbiased) training of graph neural networks in the community.

**Time Spent Reviewing:**

2

---

> ### Author Response · Authors · 2021-08-10
> **Response to Reviewer 8vwm**
>
> We appreciate the positive feedback by the reviewer!
>
> The aggregated feature can be viewed as the final hidden representation of a GCN with the weight matrix set as an identity matrix and without ReLU activation. We think an advantage of using the aggregated feature instead of the final hidden representation of GNNs in our study is that the split of subgroups does not depend on the machine learning model (i.e., it will not change when we retrain the GNNs). During this response period, we also checked splitting subgroups by the final hidden representation and observed similar decreasing patterns. This suggests that multiple measures (aggregated feature distance, geodesic distance, and distance of final hidden representation) may be used to diagnose and mitigate the biases of GNNs in the future. The aggregated feature distance and geodesic distance, as they are independent of the learning algorithm, may be more useful in understanding fairness problems in real world scenarios.

---

> > ### Comment · Reviewer_8vwm · 2021-08-24
> > **Thanks for response.**
> >
> > Thanks for the clarification on aggregated feature in the paper. I agree that aggregated feature distance and geodesic distance can help understand the fairness problem. I also notice the subgroup performance difference is consistent across these three measures. The caveat of final hidden representation is the potential de-biasing algorithm in the learning. Nice work!

---

### Official Review · Reviewer_AWUG · 2021-07-19

**Rating:** 7
**Confidence:** 3

**Summary:**

This paper proposes a novel PAC-Bayesian analysis for the generalization ability of GNNs on the node-level semi-supervised learning tasks.

**Main Review:**

This paper studies an important problem, the generalization ability of GNNs on node-level semi-supervised learning tasks. The proposed generalization bound is interesting, without relying on very strong assumptions. Experiments on several real-world datasets also demonstrate the effectiveness of the proposed theoretical analysis.

I have the following suggestions:

1. I would suggest the authors provide more details on how the generalization capability is linked to GNN fairness. What's the definition of GNN fairness?
2. Experiments are conducted on several small-scale data. Can it be easily generalized to large-scale graphs?

**Time Spent Reviewing:**

2 hours

---

> ### Author Response · Authors · 2021-08-10
> **Response to Reviewer AWUG**
>
> We appreciate the positive evaluation as well as the constructive suggestions by the reviewer.
>
> 1. The fairness measure we focus in this work is accuracy disparity, i.e., the difference of test accuracy across subgroups. At a high level, we want to understand if the GNN models systematically make worse predictions on certain subgroups of the test nodes than others. The generalization analysis provides upper bounds (with high probability) on the test errors (1 - accuracy) of different subgroups. The disparity of these upper bounds provide us theoretical insights on the accuracy disparity, which is further verified with our empirical study. In our submission, we listed some detailed connections of how the generalization bounds imply accuracy disparity of GNNs in Section 4.3. We will also make the more high-level connection clearer in the revised version of our paper. Thanks for the suggestion!
>
> 2. Following the reviewer's suggestion, we have conducted an extra preliminary experiment on a large-scale dataset, ogbn-arxiv, a node prediction dataset from Open Graph Benchmark that contains 169,343 nodes and 1,166,243 edges. We evaluated the test accuracy of GCN and GraphSAGE on subgroups divided by aggregated-feature distance. And we found that there is also a clear trend of decreasing accuracy in this much larger graph, as the subgroup becomes farther away from the training set. The test accuracies are summarized in the table below. The test nodes in Group 1 to Group 5 have increasing aggregated-feature distance to the training set of nodes. We will add more experiments in the revised version of our paper.
>
> | Model | Group 1 | Group 2 | Group 3 | Group 4 | Group 5 |
> |----------|------------|-------------|-------------|------------|-------------|
> | GCN   |  0.806    |  0.750    |  0.705     |  0.672   |  0.608    |
> | SAGE |  0.808    |  0.756    |  0.712     |  0.666   |  0.591    |

---

### Decision · Program_Chairs · 2021-09-27

**Decision:**

Accept (Spotlight)

**Comment:**

The review scores are stable towards "clear accept". Some concerns raised by a reviewer are properly addressed during the discussion period, making the paper have a unanimous consensus. Also, it makes an interesting connection with "fairness" concerning accuracy disparity, thereby opening up interesting problems along with this direction in the community.